# Learning tissue representation by identification of persistent local patterns in spatial omics data

Jovan Tanevski [1,2,3] ✉, Loan Vulliard [1,4], Miguel A. Ibarra-Arellano[1], Denis Schapiro [1,2,5], Felix J. Hartmann [4,6] & Julio Saez-Rodriguez [1,2,7] ✉

Spatial omics data provide rich molecular and structural information on tissues. Their analysis provides insights into local heterogeneity of tissues and holds promise to improve patient stratification by associating clinical observations with refined tissue representations. We introduce Kasumi, a method for identifying spatially localized neighborhood patterns of intra- and intercellular relationships that are persistent across samples and conditions. The tissue representation based on these patterns can facilitate translational tasks, as we show for stratification of cancer patients for disease progression and response to treatment using data from different experimental platforms. On these tasks, Kasumi outperforms related approaches and offers explanations of spatial coordination and relationships at the cell-type or marker level. We show that persistent patterns comprise regions of different sizes, and that non-abundant, localized relationships in the tissue are strongly associated with unfavorable outcomes.

The connection between the microanatomy of a tissue and its function is well recognized. Histological staining, immunohistochemical, and immunofluorescence panels measuring a small number of molecules are routinely used for diagnosis, tracking disease progression, assigning patients to treatable groups, and more. The advancement of spatially resolved omics technologies offers an unprecedented insight into the structure of tissues and the molecular state of their constituents[1,2], bridging biological understanding of disease to the clinical pathology practice[3,4]. This is made possible by an increased richness of the measured data, the number and diversity of distinct molecules, and the spatial resolution of tissue observations. The availability of such data prompts a revisit of the understanding of the relationship between structure and function in tissues[5,6]. Subsequently, it motivates the development of computational approaches aimed at facilitating the discovery of biological insights from complex and spatially resolved data[7–9].

The computational discovery of knowledge[10] relies on data representations, i.e., transformations of the data to a new set of variables to facilitate exploration and association to biologically and clinically relevant observations. A good representation not only facilitates the efficient and effective application of computational methods but is also explainable. That is, the models built on such representation are communicable to domain experts, experimentally verifiable, and practically deployable. In the frame of analysis of spatial omics data, there are two commonly used approaches to data representation: clustering (phenotyping) and neighborhood (niche) analysis.

Clustering assigns a label to measured spatial profiles, usually corresponding to a cell (sub)type or a functional state. By spatial profile, we here refer to the abundances of measured molecules at a specific location. This location can be captured with different resolutions ranging from tens of micrometers (a small group of cells) to a micrometer (sub-cellular resolution). While the grouping of spatial

[1]Institute for Computational Biomedicine, Heidelberg University and Heidelberg University Hospital, Heidelberg, Germany. [2]Translational Spatial Profiling Center, Heidelberg University Hospital, Heidelberg, Germany. [3]Department of Knowledge Technologies, Jožef Stefan Institute, Ljubljana, Slovenia. [4]Systems Immunology and Single-Cell Biology, German Cancer Research Center (DKFZ), Heidelberg, Germany. [5]Institute of Pathology, Heidelberg University and Heidelberg University Hospital, Heidelberg, Germany. [6]German Cancer Consortium (DKTK), Heidelberg, Germany. [7]European Molecular Biology Laboratory, European Bioinformatics Institute (EMBL-EBI), Hinxton, UK. ✉e-mail: jovan.tanevski@uni-heidelberg.de; saezlab@ebi.ac.uk

units, with or without considering spatial locations[11], is performed in a data-driven manner, the discrete labels are assigned based on known cell-type markers, functional markers, or by cross-referencing existing atlases. Consequently, the output of the clustering becomes explainable and communicable. Clustering is a first step towards representing the tissue structure. The assumption behind clustering is that the majority of the relevant information about the spatial unit is captured by its profile. Therefore, the tissue can be represented by the distribution of constituent clusters.

Neighborhood (niche) identification and analysis, although related to clustering, goes one step further. Often starting from the results of the clustering, it aims at capturing a higher-order representation of the tissue by identifying conserved patterns of interactions between clusters. Methods for neighborhood analysis adopt different approaches. Some, such as histoCAT[12], Giotto[13], and Cellular Spatial Enrichment Analysis (CSEA)[14,15], focus on the immediate neighborhood and number of interactions between pairs of cell types or functional states. The pairs of interacting cell types are identified by calculating the significance of co-occurrence within the immediate neighborhood by comparing to a null distribution of interactions derived from cell location permutations. Another group, consisting of methods such as iNiche[16], Spatial-LDA[17], and Coordinated Cellular Neighborhoods (CCN)[18,19], identifies neighborhood motifs by first representing each cell by the cell-type composition within its neighborhood and then clustering this representation again to infer higher-order motif-oriented identities. A third group of methods, such as SPACE-GM[20] and STELLAR[21], aims at supervised end-to-end GNN (graph neural network)-based learning of cell representations based on their cellular neighborhoods as defined by a cell graph[22] of the sample. Starting from the spatial profiles or initial clustering, this group of approaches relies on either additional approximate interpretation or a combination of the learned embeddings with other neighborhood analysis methods. Finally, a more recent related group of approaches including among others BANKSY[23], CellCharter[24] and UTAG[25] take the spatial profiles as input in contrast to cell-type labels. By explicitly taking into account the spatial neighborhood around each cell, they perform spatial clustering where labels do not necessarily correspond to distinct cell types, but rather to membership in a number of distinct spatial neighborhoods. Therefore, such approaches, due to the interpretation of their resulting representations, are often also considered as neighborhood analysis approaches.

Some of the previously mentioned methods are constrained by their reliance on the initial clustering of the data as a baseline representation for further neighborhood analysis. The neighborhoods in these cases are often defined based on colocalization or correlation. In the presence of multivariate or non-linear structural patterns[26], these approaches result in the identification of redundant or spurious neighborhoods which are post hoc manually relabeled or joined, before being used for downstream analyses. However, neighborhoods defined by the colocalization of cell types inherit the explainability of the baseline representation. The simplicity of the underlying assumptions means that they can be applied to data consisting of a small number of samples or even a single sample. On the other hand, GNN approaches require a large number of labeled samples for (supervised) training to reliably produce their embedded representation, which for a given tissue and a specific condition might not be available. While GNN approaches can potentially capture complex relationships from the available data, the produced embeddings are not explainable.

Here, we present Kasumi, an approach to learning a representation of tissue structures that overcomes the limitations of methods for neighborhood analysis by: (i) accounting for spatially localized patterns of multivariate, non-linear, and robust intra- and intercellular relationships; (ii) offering the flexibility to analyze neighborhoods without relying on cell-type labels; and (iii) providing an unsupervised

and explainable representation of samples as a composition of neighborhoods persistent across samples.

Kasumi aims to expand the current understanding of tissue structure and its relation to condition and function by extracting knowledge available in high-dimensional and spatially resolved omics data. To this end, we define our approach around the explainable multi-view framework for dissecting spatial relationships from highly multiplexed data (MISTy)[27]. MISTy is a general framework for extracting global relationships from spatial omics data. The output of MISTy is a set of robust relationships coming from different spatial contexts that are present across whole samples. The task commonly addressed with the MISTy framework is exploratory analysis and hypothesis generation. Kasumi extends the MISTy framework and instantiates it towards representation learning based on localized multi-view relationships. With Kasumi, we further address the task of neighborhood analysis by defining the concepts of similarity and persistence that are specific to the extended framework and the task of downstream learning by association of relationship-based representation of neighborhoods to clinical outcomes in a translational setting.

Our approach to defining neighborhoods based on persistent multivariate relationship patterns allows for more flexibility to cross scales of organization[28]. Instead of forming neighborhoods by grouping spatial units with similar profiles or similar local compositions, we define persistent neighborhoods as tissue patches that share robust relationships at different scales, i.e., different spatial contexts across the analyzed samples. In other words, instead of considering the cell as an independent unit of organization of the tissue, we consider the consistent intracellular and intercellular relationships as captured by a non-linear and multivariate predictive model as a tissue representation.

Different biological processes operate at different scales and their spatial regulation can be overlooked when searching for patterns at an inappropriate resolution[29]. Kasumi offers flexibility in the definition of the type and scale of the relationships which can capture the tissue composition (organization of cell types) or function (relationship among markers of molecular processes). Furthermore, it doesn't rely on initial clustering, but can rather use observations at the level of spatial profiles or any other available data representation.

Kasumi is scalable with regard to resolution and number of samples, and it can be used on a single sample or hundreds of samples. It is easily deployable and freely available as an open-source R package. The representation learned by Kasumi is explainable and it doesn't require the use of complex models for biologically relevant or translational downstream tasks. We demonstrate this on different cancer-related data by addressing tasks of patient stratification related to expected disease progression and response to treatment.

## Results
### Kasumi identifies persistent local patterns
Kasumi is an unsupervised multi-view modeling approach to learning representations of tissues based on localized relationship patterns in spatial omics data (Fig. 1).

Kasumi takes as input a spatially-resolved dataset consisting of one or multiple samples from the same tissue under multiple conditions. Each sample from the input data is then organized in the form of a view composition. Each spatial unit is described by its identity or functional state, forming an intrinsic view (intraview). Additional views capture the relations between each unit's properties in different spatial contexts, or represent complementary aspects of the data.

While the view composition can be defined flexibly and can be tailored to existing hypotheses or processes of interest, we focus here on a single-cell resolution and a composition of two views. In the following, all models consistently use the same view composition of an intraview and a paraview.

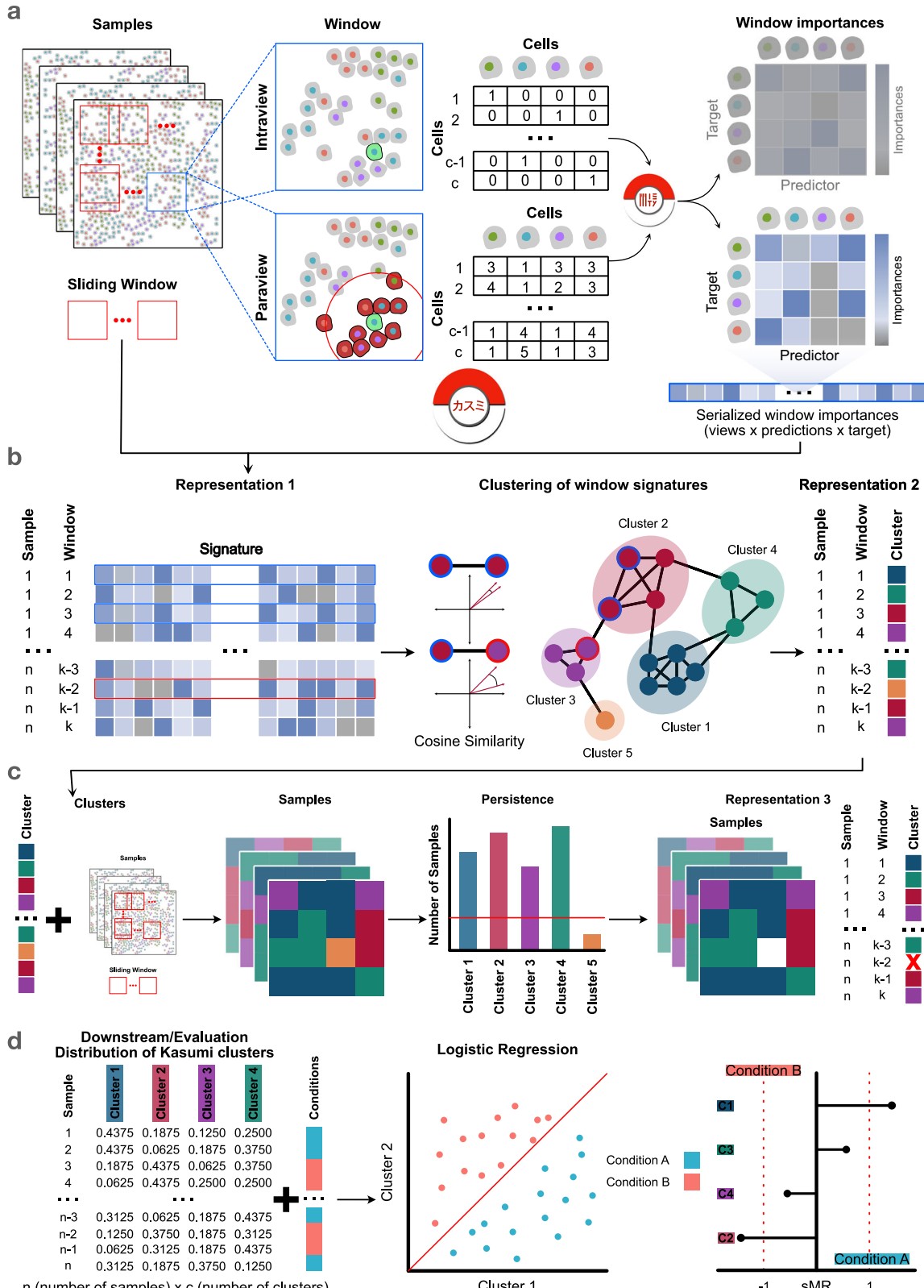

For example, given cell-type label information for each cell in a sample, the intraview simply describes each cell by its one-hot encoded cell type (Fig. 1a, top). The intraview is complemented by a broader tissue view (paraview) for each cell in the sample, capturing the cell-type composition of its neighbors (e.g., the 10 nearest neighbors in the tasks below) as a sum of the one-hot encoded vectors of the neighbors (Fig. 1a, bottom). With this view composition, we can estimate patterns of different cell types within the broader tissue structure. The resulting patterns are then defined by the importance (later defined) of the abundance of each cell type to predict the identity of a cell based on its neighbors.

Alternatively, given measurements of a number of markers for each segmented cell, the intraview describes each cell by the vector of average marker abundances in that cell. The intraview is

**Fig. 1 | Kasumi workflow and representations. a** Kasumi takes as input a number of spatially resolved measurements from tissue samples, where each spatial location is assigned a cell type (alternatively, a vector of abundances of measured markers (Supplementary Fig. S1)). Each sample is decomposed into tissue patches by sliding a window of fixed size with or without overlap. From each window, Kasumi extracts relationships coming from different spatial contexts by estimating the importance of each measured variable as a predictor of each target variable from a multi-view multivariate non-linear predictive model. The tables depict the input representation as captured by the intraview (top) and the paraview (bottom). The paraview captures the broader tissue structure by aggregating representations in the neighborhood of each cell. For example, if the intraview captures the type of each cell, the paraview captures the number of cells of a particular type among the 10 nearest neighbors of each cell. **b** The importance of the relationships per window across all samples (first representation) are used to construct a relationship similarity graph. The similarity between two windows is measured by the cosine similarity between all estimated predictor-target importances. The windows are clustered by graph community detection (second representation). The output of Kasumi is compressed, explainable, relationship-based representations of the samples, preserving relevant biological signals. **c** This is followed by cluster removal based on a persistence criterion (third representation) and mapping of the persistent cluster back to the sample windows. **d** For downstream tasks each sample is represented by the distribution of persistent Kasumi clusters at the level of a sample, capturing its local pattern composition. This representation can be used for a supervised or unsupervised downstream task. In this work, the representation together with information of the condition associated with each sample is used to train a logistic regression model to evaluate how well the Kasumi representation can predict the expected patient outcome. Finally, we provide insights into the reliance of the model on specific clusters for the prediction task in order to facilitate the explanation by focusing on condition-specific Kasumi clusters.

complemented by a paraview that captures the sum of marker abundance in the broader tissue structure, inversely weighted by the distance between the cells. The estimated relationships are patterns of intra- and intercellular predictor-target interactions of marker abundances (Supplementary Fig. S1). The Kasumi window clusters and the Kasumi representations are defined by the relationships in each window. As such, a Kasumi window with a size smaller than the size of the paraview, capturing a group of cells with patterns beyond the scope of the window might result in a low number of very similar clusters and a misleading explanation.

Within the multi-view framework of MISTy, and in turn Kasumi, these relationships are estimated by constructing a non-parametric and non-linear machine learning model in a self-supervised manner. In these models, each target variable (cell type or marker abundance) in the intrinsic view is modeled by all other variables (predictors) coming from each view independently, i.e other cell types in the neighborhood or abundances of other markers, within the cell or in the broader tissue context captured by the paraview. In each model, the estimated importance of each variable as a predictor is a quantification of its potential relationship with the target variable in the context of each view. The estimated importances per target are further standardized to have zero mean and unit variance, to render values comparable across predictions. A global representation of the sample is a selection of the set of relationships with the highest importance. For example, one can select only the predictor-target relationships one standard deviation above the mean importance per target and view. As standardized importance estimates are also comparable across samples, aggregating the importances across samples allows for the selection of the most robust predictor-target relationships overall.

Within a single sample, the estimated patterns of relationships between cell types go beyond simple co-localization. They are multivariate, can be non-linear and are determined by the form of the underlying predictive model, while still being explainable through the estimated importance of each predictor-target relationship. For example, as predictive models we use Random Forests[30] and the importances are estimated as a function of the total reduction of variance of the target as the result of splitting by each of the variables in all constituent trees (see Importance estimation).

Since the above-described modeling occurs at the level of a whole sample like in the context of the global MISTy instance, the resulting representation captures a global view of relationships present in the tissue and not the local heterogeneity of tissue sub-structures and their potentially diverse functions.

To address this issue, Kasumi learns a representation based on tissue patches sharing relationship patterns that define local neighborhoods (Fig. 1a). Representing whether such patterns are global or local better accounts for spatial and molecular heterogeneity within samples.

A neighborhood is a regularly shaped, spatially contiguous region (window) of the sample. The sliding window is defined at the level of

the view composition, i.e., across all spatial contexts captured by the view composition at the same time.

The representation of window $k$ then is the ordered set of estimated standardized local importances of the relationships $\{M_{j,i}^{(v,k)}\}$ given local models combining all views $v$ for a target $i$, predictors $j$ in a window $k$:

$$\mathbf{Y}_{.,i}^{(k)} = \alpha_I^{(k)} + \alpha_0^{(k)}(F_0^k \circ W)(X, s, o, k, \bar{\mathbf{Y}}) + \sum_v \alpha_v^{(k)}(F_v^k \circ W \circ G_v)(X, s, o, k, \bar{\mathbf{Y}}, T)), \quad (1)$$

where $\mathbf{Y}_{.,i}$ is the vector of all observations of variable $i$ in the intrinsic view, $\bar{\mathbf{Y}} = \mathbf{Y}_{.,\forall q \neq i}$, $F_v^k$ are the trained machine learning models predicting values $\mathbf{Y}_{.,i}$ from variables in view $v$, W is a function that subsets units of view $v$ that belong to the sliding window $k$ of size $s$ and percentage of window overlap $o$, G are domain-specific functions that transform the intrinsic view based on spatial context X (for example, aggregation of the representation of the 10 nearest neighbors or distance-weighted sum of variables like described in Section 2.4) and functional context $T$ (for example, estimated pathway activities or selection of variables representing ligands or receptors) and $\alpha$ are parameters of the $L_2$ regularized late fusion linear meta-model that is trained on the predictions from the independently trained view-specific models.

Each sample is then represented by its compressed and relationship-oriented form of a matrix of estimated importances M with the same number of rows as the total number of valid windows and a number of columns equal to the number of all predictor-target feature combinations with importance larger than zero, comprising the first representation of Kasumi (Fig. 1b, left, see also section Kasumi importance signatures). To trade off computational time with sample coverage, we consider a window overlap of 50% by default (see Computational complexity). To avoid overfitting, we discard windows that contain fewer cells (spatial units) than features in the intraview.

In general, given data from multiple samples, it is difficult to compare directly the abundances and cell-type distributions across samples without further data manipulation and integration. This is mainly due to sampling, batch or technical effects. In contrast, the relationship-based representation is invariant to these effects and can be directly compared.

Therefore we can define the notion of similarity in the frame of local relationship-based representation of neighborhoods within and across samples. Based on the similarity structure, we further define consistent relationship-oriented clusters of windows. Namely, since the importances are estimated per target from the intrinsic view when comparing the importance-based representation within and across samples we are interested in their relative values. To establish a similarity structure, we define and cluster a graph of similarities of all windows across all samples. Each node in the graph represents a window described by the Kasumi importance signature, i.e., the set of all estimated predictor-target relationships from that window with

importance larger than zero (subsection Kasumi importance signatures). The nodes in the graph are connected by edges with weights equal to the cosine similarity between the node representations (Fig. 1b). The edges of the fully connected graph are filtered based on a similarity cutoff. The Kasumi clusters are then determined by Leiden community detection[31] (see Predictive tasks and sensitivity analysis).

Labeling each window with the community it belongs to leads to the second compressed representation of samples as a vector of discrete window identities based on shared signatures of relationships across samples coming from the same tissue and condition (Fig. 1b, c).

The existence of shared local structural (cell type) and functional (marker) patterns opens up a venue for exploring higher-level organization and empirical models of tissues. To define the building blocks of such representation as inputs to downstream tasks, as well as further reduce the complexity of the representation, we regularized the description by retaining only clusters that match a persistence criterion (Fig. 1c). Local patterns are meaningful for reasoning in terms of knowledge discovery and application to downstream tasks if they are consistently present across samples. The multiple independent observations of the same pattern within a subset of tissues in or across conditions adds to the relevance of the neighborhood to that tissue or condition. It also reduces the chances to include neighborhoods specific to a single sample that add noise and bias to downstream analyzes. To this end, we retain clusters that are present in at least 10% or at least 5 samples. The clusters not fulfilling this criterion are not considered further. By removing noise and sample-specific information, we also reduce the complexity of the representation, leading to the third and final representation of the samples by persistent local patterns (Fig. 1c).

The output of Kasumi consists of the three layers of representation and the summary model statistics for each target, view, and sample. All Kasumi representations are at the level of a window (neighborhood) and are comparable across samples. Each sample is represented as a composition of windows. Each window is represented first as a vector of importances of length equal to the total number of all underlying predictor-target relationships across views, second as a single cluster label, consistent across samples, and third, as the same cluster label if not filtered out based on the persistence criterion. Windows that are filtered out are simply removed from the representation and not considered further. In the following, we used the third layer of representation by persistent local patterns, aggregated at the level of a sample and represented as the distribution of Kasumi clusters (Fig. 1d), as inputs to the predictive tasks and the other two layers to support the explanation.

We learn Kasumi representations of samples coming from three different cohorts of oncological patients, measured respectively with three different spatial proteomics technologies. We learn neighborhood level sample representations starting from two initial cell level representations of the samples—cell-type label and marker abundance level information for each cell in each sample with each of the three cohorts. To evaluate the proposed approach we compare the performance of the learned representation to several baselines and related approaches on the task of patient stratification. We measure the performance given a ground truth of follow up observations of disease progression and response to treatment (Fig. 1d).

### Kasumi extracts robust patterns associated with clinical features

The representation generated by Kasumi is data-driven and unsupervised. It can be used for different downstream tasks. To demonstrate the ability of Kasumi to retain biologically relevant information while offering a simpler, lower-dimensional representation of the tissue, we evaluated it on different clinically relevant predictive tasks. For these tasks, we use the distribution of Kasumi clusters as sample representation (Fig. 1d).

We considered highly multiplexed and spatially resolved data consisting of samples of ductal carcinoma in situ (DCIS)[15] measured by multiplexed ion beam imaging (MIBI)[32], samples of cutaneous T cell lymphoma (CTCL)[19] measured by co-detection by indexing (CODEX)[16] and breast cancer (BC)[33] samples measured by imaging mass cytometry (IMC)[34]. We chose recent studies that shared their data and showcased different types of neighborhood analyses. We focus on benchmarking Kasumi together with explainable approaches that can perform at the level of a single sample and can reliably scale up to a large number of samples, namely the CSEA[15] and CCN[19] methods.

As a global composition-based representation without taking the spatial context into account, we considered the simple cell-type distributions per sample. By defining a global, non-spatial composition as baseline we aim to show that there are no significant differences between the groups of patients (responders vs non-responders, progressors vs. non-progressors) when the spatial context of the data is not taken into account. Consequently, the improvement of performance, given the different neighborhood analysis approaches, is due to considering the spatial organization of the cell types and the relationships between them within the different spatial contexts.

To demonstrate the added value of considering local patterns of relationships instead of global relationships per sample, we also ran MISTy[27] using the same view composition as Kasumi. Since MISTy doesn't provide any neighborhood information, as representation we consider the importances of the relationships across all views for each sample. This also demonstrates that the global relationship-centric representation offered by MISTy can already complement the baseline approach, solely relying on cell type composition.

Finally, in order to directly estimate the value of the local relationship-based representation of Kasumi relative to a composition based neighborhood representation, we implemented a clustering of composition of sliding windows across samples (WCC). Instead of representing each Kasumi window by the estimated relationships, we represent it by the normalized vector of composition of cell types within that window. The resulting representation is comparable across windows and samples and can be further considered as a local composition-based baseline. As for Kasumi, we first remove the windows that contain fewer cells than the total number of cell types. We next cluster the windows and represent each sample by the frequency of its window clusters in the same way as for Kasumi. To ensure direct comparison to Kasumi, we set the window size and the number of clusters to match the best performing Kasumi run. Any performance gain of Kasumi in addition to the Window Composition Clustering (WCC) can therefore be attributed to the relationship-based representation capturing relevant interactions beyond composition.

We evaluated the performance of simple logistic regression models on the tasks of predicting progression to breast cancer from the DCIS samples ($n = 58$, 44 non-progressors, 14 progressors), predicting the response to PD-1 inhibition in pre-treatment CTCL samples ($n = 29$, 14 responders, 15 non-responders) and sensitivity to post-operative treatment with hormone therapy in Estrogen and Progesterone receptor positive, HER2 negative BC samples ($n = 30$, 15 sensitive, 15 resistant). The ability to use simple models based on a representation without further transformation suggests that the latter encodes relevant biological information and is easily explainable.

Based on the performance estimated by 10-fold cross-validation and summarized by the area under the receiver operating characteristic curve (macro-AUROC) the Kasumi representation outperforms the global composition representation of cell-type distribution on all tasks, underscoring the added value of spatial information (Fig. 2a). Kasumi also clearly outperforms all other representations generated by related approaches (Fig. 2a), even with different choices of window sizes (Fig. 2b). The local composition-based approach WCC shows better performance than related tasks than related approaches on two out of three tasks, while also outperforming the global composition on the

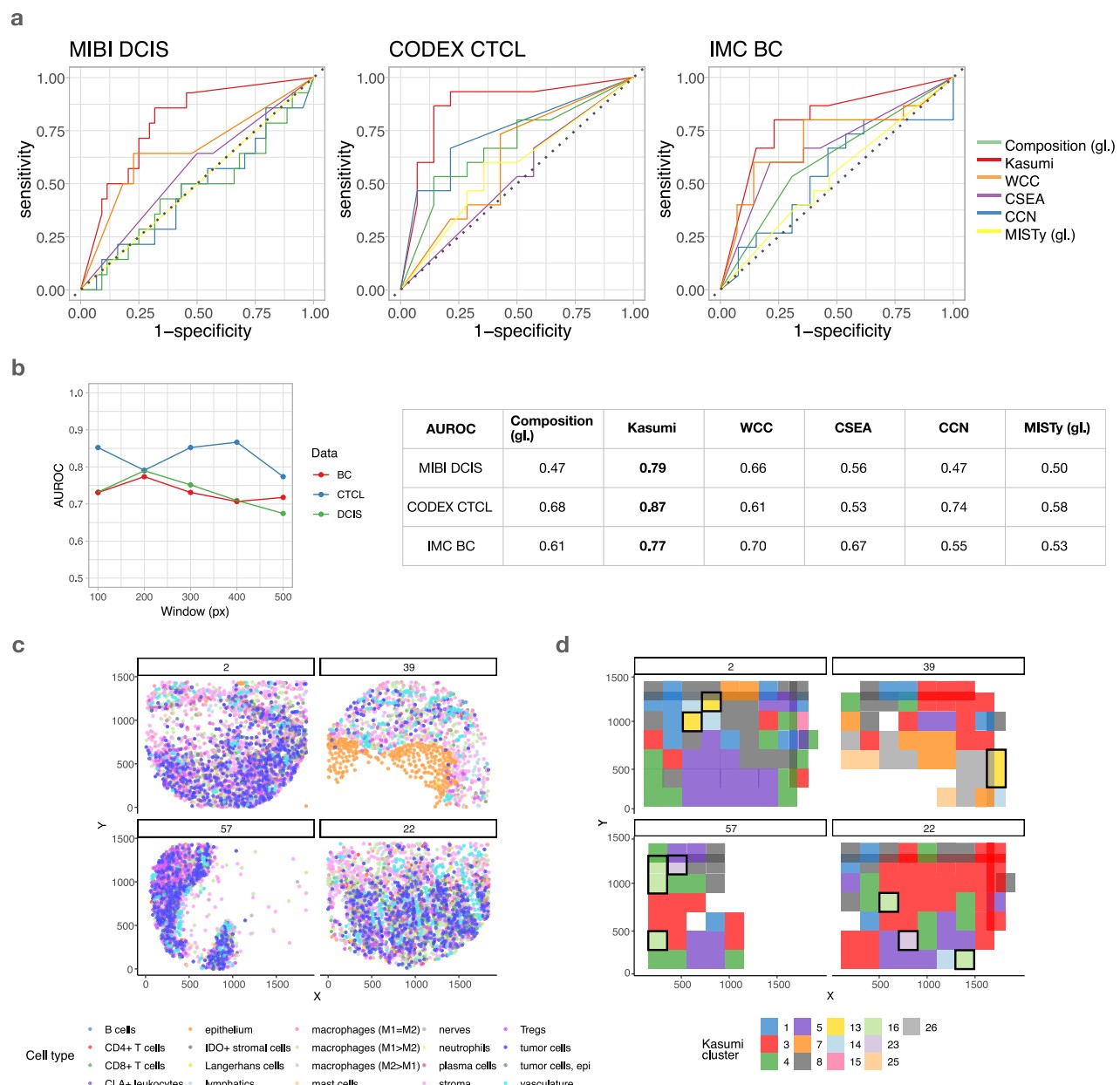

**Fig. 2 | Performance of the Kasumi representation in a cell-type based scenario on downstream tasks. a** Kasumi outperforms a global composition-based representation of sample cell-type distribution, a global relationship-based representation (MISTy), a local window composition clustering approach (WCC) and related local neighborhood analysis approaches on tasks of predicting disease progression and response to treatment. Bold values in the table indicate the AUROC of the best performing approach per dataset. The abbreviation gl. stands for global representation in contrast to the local representation generated by the other approaches. **b** The window size affects the performance of Kasumi. Larger window sizes decrease the performance, pointing to the importance of local patterns. **c** Location of cell centroids labeled by cell type (cell state) as represented in the original dataset. Top—two samples of responder patients from the cutaneous T cell lymphoma (CTCL) data; Bottom—two samples from non-responder patients. **d** Resulting Kasumi representation for the samples shown in c. Each window covers multiple cells and the label (color) assigned to the window is based on the clustering of the relationships extracted from that window. Kasumi offers a spatially-aware, compressed but explainable representation that preserves relevant biological information. Highlighted are the windows of the Kasumi representation. The highlighted windows are taken from the results shown and further explained in Fig. 3, identified as the most important neighborhoods to distinguish between responders and non-responders. The missing windows (patches) in the Kasumi representation of the samples compared to the denser regions shown in (c) are a result of applying the persistence criterion. Source data are provided as a Source Data file.

same two tasks. Instead of providing labels at the level of a single cell, both Kasumi and WCC compress and cluster the tissue at the level of patches. Based on the results, we conjecture that the reduction of the size of the representation of the sample by considering patches instead of cell level labels leads to the improvement in performance. We further compared the clustering produced by WCC and Kasumi (Supplementary Fig. S7), and observed a low similarity in terms of the Adjusted Rand Index (ARI). The mean ARI across samples is 0.19, 0.25 and 0.21 for DCIS, CTCL and BC respectively. The higher performance of Kasumi across all tasks relative to all other composition-based approaches highlights the benefits of its relationship-based representation.

The performance of Kasumi across tasks consistently decreases with the increase in window size after a peak performance value (Fig. 2b), pointing to the importance of capturing the local

heterogeneity of tissues. Note also that the optimal choice of window size is dependent on the task and the scale of the important biological processes for that task. For the optimal window choice, the learned Kasumi representation consisted of 33, 16 and 9 unique persistent window clusters for the DCIS, CTCL and BC datasets respectively. Not applying the persistence criterion leads to poorer results, despite a higher number of clusters (57, 28 and 28 for the DCIS, CTCL and BC datasets respectively). The performance decreased compared to the representation applying the persistence criterion (AUROC of 0.53, 0.66 and 0.53 compared to 0.79, 0.87 and 0.77 for the DCIS, CTCL and BC datasets respectively). An example of the resulting Kasumi compressed representation obtained from a cell-type based sample representation (Fig. 2c) is shown in Fig. 2d. The cell-type composition, WCC clusters and Kasumi clusters for all samples are shown in Supplementary Figs. S2 and S3. The effect of the persistence criterion can be observed as missing windows in the Kasumi representation of the samples despite including a sufficient number of cells. In the following section, we also detail the selection process and the properties of the highlighted windows that are most relevant to distinguishing between responders and non-responders.

In general, local representations outperform global representations, and taking the spatial context into account always improves the performance on the selected tasks. We further illustrate the explainability aspect of the Kasumi representation, providing insights into the neighborhood composition and the local relationship patterns.

## Kasumi offers comprehensive explanation

The representation generated by Kasumi not only performs well in predictive tasks, but it is also explanatory. Given the predictive tasks, we can contextualize the Kasumi representation by assigning each cluster a relevance value for the task at hand by measuring the signed Model Reliance (Fig. 3a). The signed Model Reliance (sMR) value estimates the relevance of each cluster[35], while the sign indicates the contribution of the cluster to the prediction of one of the two classes (see Methods).

We can explore and explain the cluster-specific models. The Kasumi representation is task-agnostic, i.e., generated in an unsupervised manner, building upon relationships extracted from the available measurements in a self-supervised manner to represent each sample as a composition of clusters, each capturing a set of window-centered relationships. Every cluster can be explained by three aspects. First, we consider the performance of the cluster-specific models in predicting cell identity from different views. The predictive performance of the models acts as a proxy for the amount of information captured from the data. In particular, we express the performance by the variance explained by the model ($R^2$) for each target and the gain of variance explained when considering the spatial context (i.e., all views). We next estimate the cluster-specific importance of each predictor-target relationship coming from each view. All of the information needed to generate the explanations is already available in the Kasumi output and can be obtained by backtracking the relationship from the third to the first level of representation, i.e., from cluster label to performance and importance relationships.

In all tasks, the simple view compositions we considered consist of two views – an intrinsic view (intraview) and a broader tissue view (paraview). In the case of one-hot encoded cell types, we don't model the intraview as such a prediction is trivial. The gain in variance explained is therefore equal to the total variance explained, and the contribution is limited to the broader tissue context (Fig. 3b top). Each cluster is then explained by the estimated importances of the predictor-target relationships (Fig. 3b). To remove spurious relationships and minimize the number of false positives in our explanation, we removed targets with low explained variance and estimated importances below one standard deviation above the mean ($M \geq 1$). For completeness, all estimated importances above zero are shown in Supplementary Fig. S4.

The underlying model is multivariate and non-linear and therefore the Kasumi relationships should not be interpreted as linear or in a pairwise predictor-target manner. The explanation of the results should be at the level of estimated importances. However, to improve the explainability of the results, an approximation of the sign of the relationship between the cell types (co-localization or avoidance) can be estimated by the correlation between the presence of each target cell type and the abundance of the important predictor cell types in the broader tissue structure across all windows belonging to the same Kasumi cluster. Note that low correlations of predictor-target relationships with high estimated importances are indicative of non-linearity or heterogeneity of the underlying relationships. In particular, the sign of the interaction between important relationships might be different in different windows, which in turn can be reflected as a low overall correlation. High correlations are indicative of a consistent linear relationship between a predictor and a target.

For example in the best-performing case, analyzing the results of the task pertaining to PD-1 inhibition treatment in CTCL, the model is highly reliant (sMR < −2) on clusters 23 and 16 to determine if the sample belongs to the group of non-responders, and on clusters 4 and 13 (sMR > 1) to responders (Fig. 3a also shown on samples in Fig. 2d). From these clusters 16 and 4 capture patterns of negative relationships between the tumor, stroma, and vasculature. The difference in patterns between the clusters is the presence of a positive relationship between the tumor and macrophages (M1 > M2) in cluster 4 that have been shown to be associated with favorable outcome due to their anti-tumorigenesis role[36]. In cluster 13, associated with responder outcome, we also observed tumor-associated macrophages, in particular M1 = M2 and M1 > M2, to be positively interacting with the stroma. On the non-responder side a pattern of avoidance of M1 > M2 macrophages and tumor cells is also captured by cluster 23. In summary, patients with low immune activity in the tumor microenvironment are unlikely to benefit from PD-1 inhibition. This exemplifies how the explanation of the Kasumi models can offer complementary perspectives on multiplexed imaging datasets and guide hypothesis generation.

The description of the interactions supporting each cluster can be complemented by its abundance and distribution across samples. How often and where a cluster is found informs us about the biological process it may represent. We observe that the frequency at which clusters are found varies greatly, demonstrating the ability of Kasumi to model phenomena at different scales (Fig. 3c). For instance, cluster 4 is found in most samples although 1.6 times more common in samples from patients who responded to the immunotherapy. Cluster 16, on the other hand, is associated with non-responders although present in only a small number of windows. The contrast between cluster 4, describing a common tumor-related pattern, and cluster 16 highlights the relevance of both global, cancer-related patterns and local microenvironment structures. Similar observation can be made for clusters 13 and 23. The cluster frequency is independent of the model reliance for these clusters, regardless of whether they are associated with positive or negative outcomes. Clusters 23, 16, and 13 have relatively low frequency but high relevance for the predictive task, while cluster 4 has high frequency but is also found to be relevant. Clusters that are found in multiple windows per sample (Fig. 3c) are also spatially autocorrelated (Fig. 3d), i.e., show spatial contiguity. The spatial contiguity of Kasumi clusters is significantly stronger than expected from a random uniform distribution and systematically points at processes restricted to local tissue regions.

In addition to offering a more comprehensive explanation, Kasumi reduces the complexity of representation (number of clusters) compared to the number of cell types (Fig. 2c vs. d). For simplicity and in order to show the amount of redundancy of the representation, the logistic models were not regularized. Clusters with an estimated absolute value of sMR above 1 are considered to be relevant for the

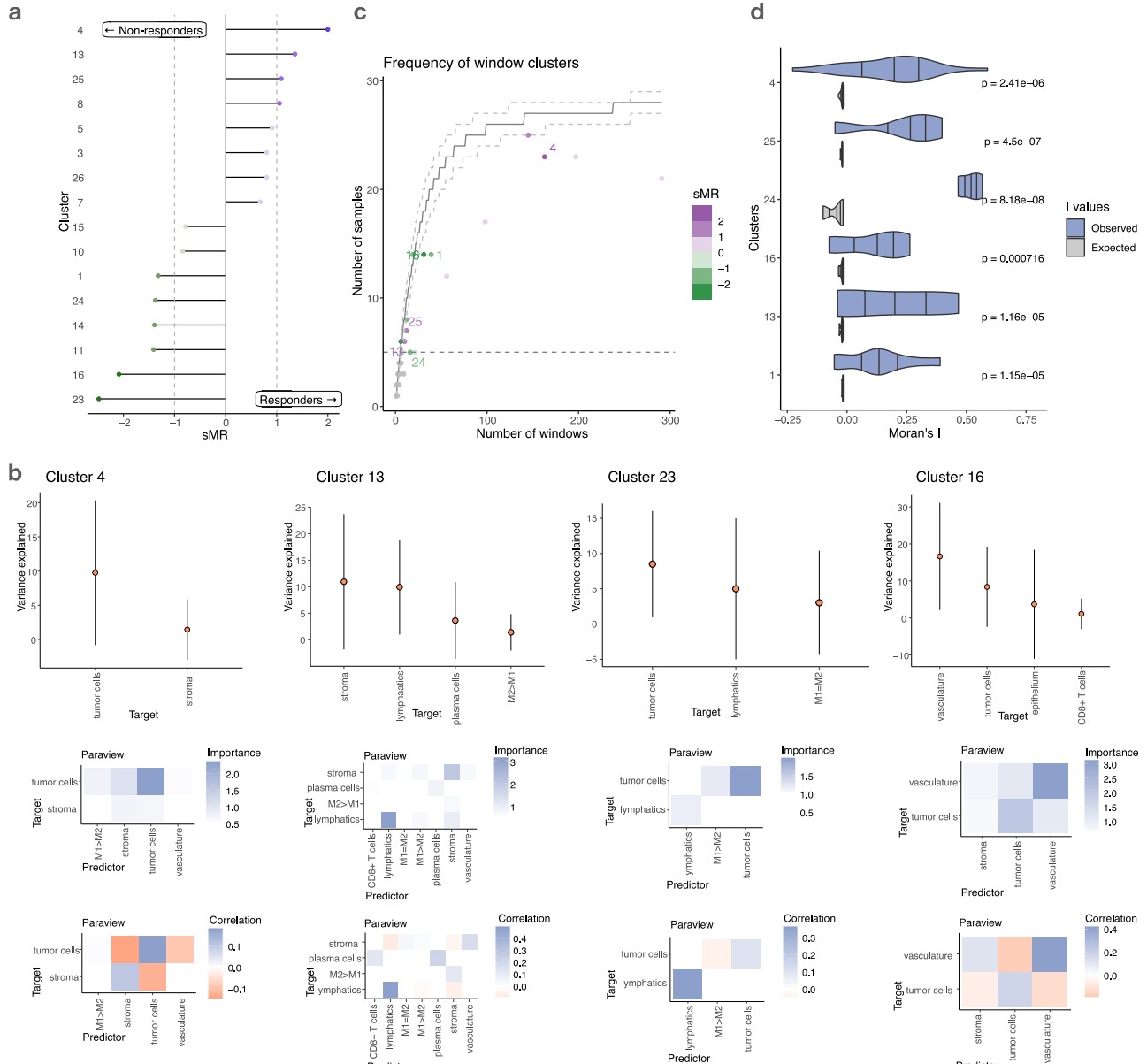

**Fig. 3 | Explainability of the Kasumi representation. a** Signed reliance of a simple logistic regression model on the frequency of individual Kasumi clusters as sample representation for the task of distinguishing responders from non-responders in the cutaneous T cell lymphoma (CTCL) data. **b** Distribution of variance explained per target cell type (cell state) across all windows from the same cluster (top, targets with variance explained less than 1% are not shown). Points represent the mean and the bars represent standard deviation across windows labeled with the corresponding cluster ($n = 163$ for cluster 4, $n = 10$ for cluster 13, $n = 6$ for cluster 23, and $n = 31$ for cluster 16). Estimated average importances of relationships explaining the cluster (middle), and average correlation between the predictor and target of important relationships across all windows coming from all samples (bottom). **c** Number of total observed windows across samples for independent Kasumi clusters. The horizontal dotted line denotes the persistence threshold to exclude clusters. The solid line represents the median simulated number of samples for a cluster found in a given number of windows, as modeled by a multivariate hypergeometric distribution. The dashed lines depict the 80% confidence interval of this distribution. Relevant Kasumi clusters are consistently found less frequently than expected. **d** Moran's I values for a sample of relevant clusters. The *p*-values of the positive deviation from I values expected under the null hypothesis of no spatial autocorrelation were computed by a one-sided test of the z-score of the observed I. The *p*-values are Bonferroni-corrected per cluster across all samples with at least two windows matched to the cluster, and Benjamini-Hochberg corrected across clusters. Source data are provided as a Source Data file.

task and can be further ordered by importance. The clusters with an absolute value of sMR below 1 are not relevant to the task or contain redundant information and can be excluded from further interpretation or analysis.

### Kasumi can model quantitative cell features

Many spatial analyses are solely based on cell types inferred from lineage markers, disregarding other information typically available, such as the abundance of functional markers. Kasumi, however, is not limited to cell-type-level input and can model all available measurements. The flexible definition of view composition as input allows for cell-type-agnostic analysis directly at the level of the observed spatial profiles.

To broaden the definition of the state of each cell to include functional aspects, the intrinsic view can alternatively describe each cell by the mean or median abundance of the observed markers. The broader tissue context (paraview) for each cell then corresponds to the relative-distance-weighted sum of abundances. The paraview of each

cell is calculated by the sum of marker abundances weighted by the radial basis function $w = e^{-\frac{d_{ij}^2}{l^2}}$. Here, $d_{ij}$ is the Euclidean distance between cells $i$ and $j$ and $l$ is a parameter that, for all tasks below, we set to $l = 100$ pixels.

In this manner, Kasumi facilitates a more consistent relationship-oriented representation across spatial contexts. Instead of estimating relationships between predefined cell types, here each window is also characterized by the intrinsic relationships that are preserved within the window, offering a different perspective and generating alternative hypotheses to explore.

On the same datasets as before, but starting from a spatial profile represented by the abundance of markers per segmented cell, we benchmark Kasumi together with several baselines and related approaches to clustering and neighborhood analysis (BANKSY[23], CellCharter[24] and UTAG[25]) on the downstream task of patient stratification given ground truth observations of progression and response to treatment. These methods are also able to leverage quantitative cellular features rather than inferred cell types. The parameters of the related approaches were optimized for performance on the downstream tasks following guidelines in the original manuscripts and implementations (see parametrization of related approaches).

As a global composition-based representation baseline with no spatial context, we considered the sum of abundances per marker relative to the total abundance in the sample. As for the previous set of analyzes, with the global marker composition we aim to show that there are no significant differences between the groups of patients and that the gain in performance is due to the local spatial distribution of marker abundances and the relationships within the different spatial contexts.

Our Window Composition Clustering (WCC) approach is also applicable at the level of observed spatial profiles. In this scenario, each sliding window is first represented by the normalized abundance of markers across cells within it. As previously, clustering this representation across all windows and samples results in sample representation based on composition of window clusters. The size of the window and the number of clusters were chosen to match the best Kasumi run. Therefore its representation and performance are directly comparable to Kasumi. Comparing Kasumi and WCC provides an estimate of the performance gained when considering relationships instead of composition at the level of each window.

The performance of the Kasumi representation on the same tasks, given spatial profiles (marker abundances) at input remains consistently high (Fig. 4a). For the optimal window choice the learned Kasumi representation consisted of 4, 12 and 15 unique persistent window clusters for the DCIS, CTCL and BC datasets respectively. Here, as in the cell-type-based scenario, bypassing the persistence criterion results in a representation based on a higher number of clusters and a significant reduction in performance (AUROC of 0.63, 0.59 and 0.62 compared to 0.72, 0.78 and 0.77 for the DCIS, CTCL and BC datasets respectively).

As in the cell-type-based analyses, local representations outperform global representations. Here, a global representation was obtained by running MISTy at the level of the whole slide with the same composition as Kasumi. Learning a marker-based relationship representation of tissue patches, Kasumi outperforms the global composition and the performance of the global MISTy approach. Furthermore, considering the underlying relationships rather than the baseline of their average values also appeared beneficial. The global MISTy approach outperforms the global composition on the MIBI DCIS dataset, as well as other related methods.

Kasumi outperforms the related approaches on two out of the three datasets. CellCharter performs better on the CTCL dataset (AUROC 0.79 vs. 0.78). BANKSY shows comparable performance on the BC dataset (AUROC 0.74 vs. 0.77). All approaches perform worse

than Kasumi run at the cell-type level. While Kasumi run at the level of marker abundances results in slightly lower performance than its cell-type counterpart, it remains consistently close. The high performance of WCC on marker abundances on two out of the three tasks points to the relevance of the marker abundance based representation of local tissue patches as predictors of outcome. The similarity of the Kasumi and WCC clustering measured by the Adjusted Rand Index is 0.61, 0.23 and 0.14 for the DCIS, CTCL and BC datasets respectively. The distribution of ARI per sample for the different datasets is shown in Supplementary Fig. S7.

The comparable performance of Kasumi on these two tasks points towards the flexibility of Kasumi to also capture composition-based information only at the level of marker abundances on par (CTCL), or better than related neighborhood analysis approaches (DCIS). In the presence of information beyond the composition of marker abundance, like in the case of the BC dataset, Kasumi significantly outperforms WCC.

The same task-oriented analyses and explanations can be applied in the context of explaining the performance and the marker relationships captured by the Kasumi representation beyond composition. Figure 4b shows the sMR values of the Kasumi clusters for the task of predicting sensitivity to treatment with hormonal therapy in breast cancer samples.

Interaction patterns within clusters show local (window-scale) cell-type marker but also functional marker relationships that are consistent and shared in up to 25 out of 30 samples (Fig. 4c). Here as well, Kasumi is able to highlight local organization at different scales and frequencies. Furthermore, across all Kasumi clusters, the windows assigned the same cluster label show significant spatial autocorrelation (Fig. 4d). We observe that some markers, and by extension the biological processes they contribute to, are more influenced by their spatial context than others. This includes both common lineage markers, such as CD3 and CD45, and markers more directly related to cellular state and function, such as CAIX and HER2.

In Fig. 4e we explore in more detail features underlying Clusters 37 and 5, found to be most relevant for the predictive task. Of particular interest, Cluster 37 in Kasumi's output was the most important for predicting resistant samples, and was found only in isolated or pairs of adjacent sliding windows (Supplementary Fig. S6c). Here, the clusters are not identified based on marker intensities alone but rather based on the relationships between markers with high estimated importance within the cells or in the broader tissue structure (Fig. 4e). This points towards the relevance of cellular and spatial dependencies in predicting treatment response. For both Cluster 37 and Cluster 5 the majority of the variance can be explained by the relationships identified in the intraview. Additional gain of variance explained can be assigned to relationships from the paraview. For each cluster, we selected only the targets for which adding the spatial context resulted in an increase of explained variance, in order to focus on spatially-regulated relationships. The variance explained per target marker for a model containing only an intraview compared to a model with both intraview and paraview as well as the relative contribution of the views for explaining the variance of the selected targets are shown in Supplementary Fig. S8. To remove spurious relationships and reduce the number of false positives we further removed interactions with estimated importances below one standard deviation for the intraview and 0.5 standard deviations for the paraview. All estimated importances in the intraview and paraview with values above zero for clusters 37 and 5 are shown in Supplementary Fig. S8.

We also detected fewer marker relationships within cells (intraview) in Cluster 37 than in Cluster 5, consistent with a loss of regulatory mechanisms in advanced and more resistant tumors. Of note, an immune pattern involving CD44, CD45, and CD68 was found in the intraview and further regulated in the paraview and was present in both clusters, highlighting an immune origin to both clusters. In the

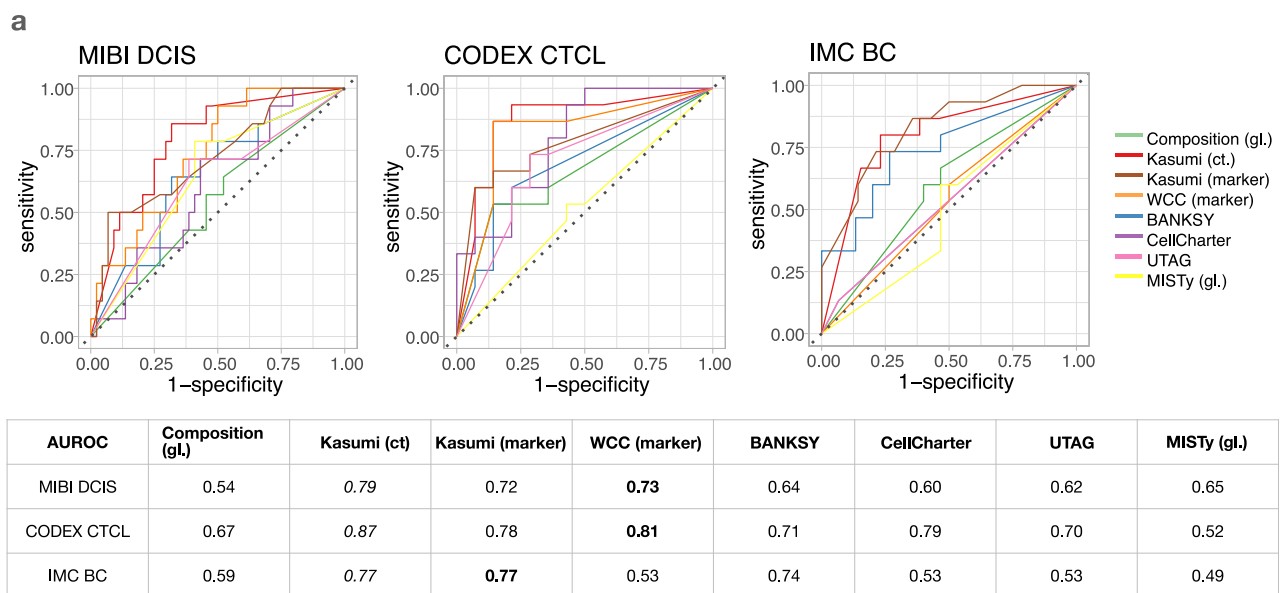

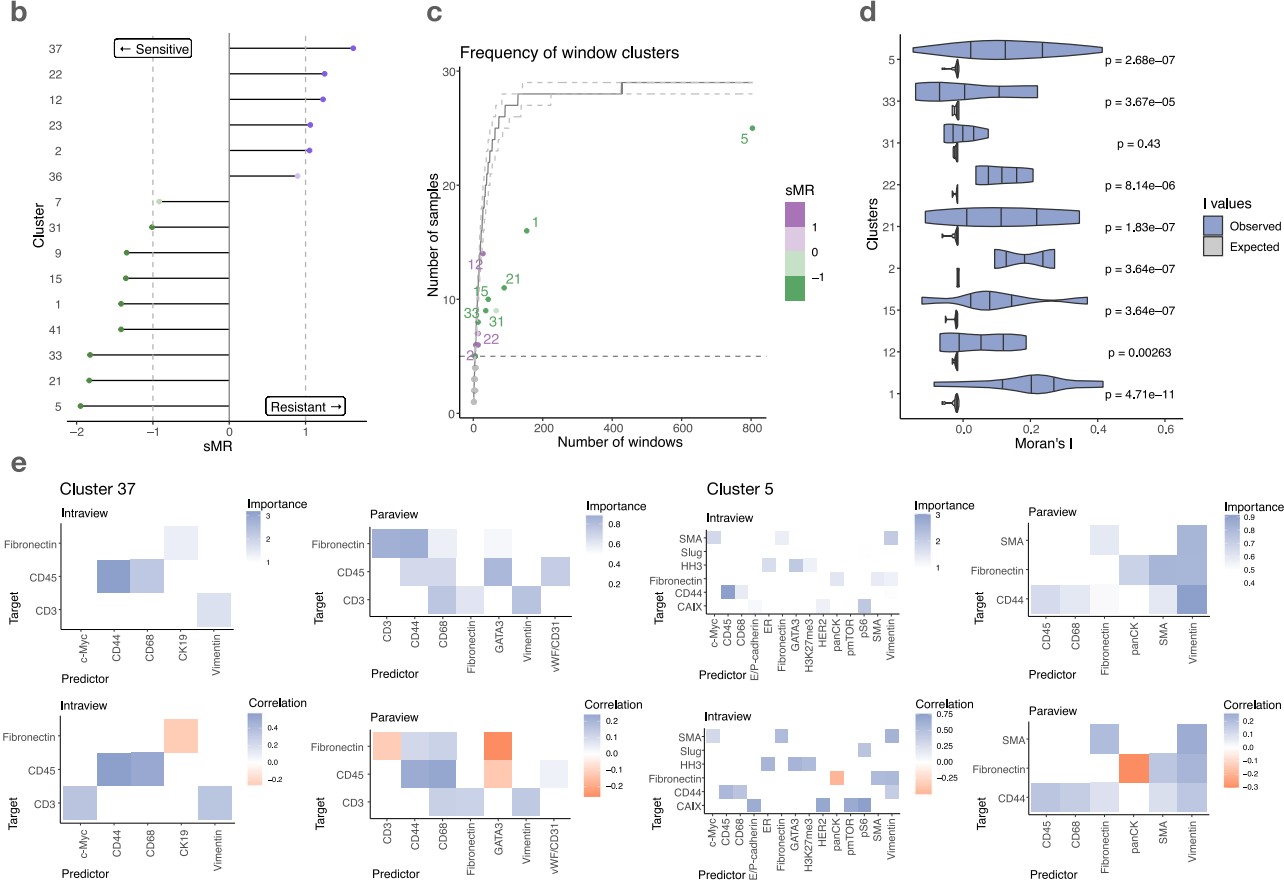

broader tissue structure (paraview), we noted an exclusion pattern between CD45, Fibronectin and GATA3. Kasumi was able to identify a 3-way pattern involving the immune, stromal and tumoral compartments as well as their relative spatial organization. By comparison, BANKSY captured a pattern involving GATA3 positive cancer cells (cluster 1, Supplementary Fig. S6a), but from its output it is not apparent how they are organized with respect to other cell classes. GATA3 is highly expressed in healthy epithelial cells lining ducts in breast tissues and are known to form a positive feedback loop with the estrogen receptor (ER)-alpha expression[37]. Hormone-receptor positive tumors are also known to have immunosuppressive

microenvironments and low immune infiltration[38]. Our observation suggests that this regulation is not present in the whole tumor sample but acting locally within tumors with immune deserts. GATA3 expression is typically a marker of good prognosis. By its association with ER, it can be indicative of a response to hormonal treatments. However, this is not the case for the five samples in the cohort that contain cluster 37. The high fibronectin intensity relates to a dense stromal compartment that may prevent immune infiltration and the availability of ER. In such cases, the hormone treatment can potentially benefit from coupling with an intervention on the tumor-immune microenvironment. Fully resolving the mechanism at play would,

**Fig. 4 | Performance and explainability of the Kasumi representation in a marker-abundance based scenario on downstream tasks. a** Kasumi performance compared to a global composition-based representation of sample marker abundance distribution, a global relationship-based representation (MISTy), a local window marker-composition clustering approach (WCC), and cell type (cell state) based Kasumi representation on tasks of predicting disease progression and response to treatment. Bold values in the table indicate the AUROC of the best performing approach per dataset. Italic values indicate the AUROC of the best performing approach in the cell-type based scenario (abbreviated as ct.) for reference. The abbreviation gl. stands for global representation in contrast of the local representation generated by the other approaches. **b** Signed reliance of a simple logistic regression model on the frequency of individual Kasumi clusters as sample representation for the task of distinguishing resistant from sensitive samples in the breast cancer data. **c** Number of total observed windows across samples for independent Kasumi clusters. The horizontal dotted line denotes the persistence threshold for the exclusion of clusters. The solid line represents the median simulated number of samples for a cluster found in a given number of windows, as modeled by a multivariate hypergeometric distribution. The dashed lines depict the 80% confidence interval of this distribution. Relevant Kasumi clusters are consistently found to be less frequent than expected. **d** Moran's I values for a sample of relevant clusters. The $p$-values of the positive deviation from I values expected under the null hypothesis of no spatial autocorrelation were computed by a one-sided test of the z-score of the observed I. The $p$-values are Bonferroni-corrected per cluster across all samples with at least two windows matched to the cluster, and Benjamini-Hochberg corrected across clusters. **e** Estimated average importances of relationships explaining the cluster and average correlation between the predictor and target of important marker relationships across all windows from all samples in two spatial contexts: intrinsically (within each cell) and in the broader tissue structure. Source data are provided as a Source Data file.

however, require additional mutational status information, as GATA3 mutations have been implicated in resistance to hormonal treatment by uncoupling the hormone signaling[39].

Treatment sensitive samples did not display such immune deserts with high GATA3 or fibronectin-expressing cells (Supplementary Fig. S5c). BANKSY identified duct-like structures (Supplementary Fig. S5a), however they did not appear more relevant for the treatment prediction task than the clusters identified by Kasumi. Cluster 5, on the other hand was found in high frequency in most samples. While it shows a similar immune patterns as in Cluster 37, it is complemented with a well-coordinated organization of multiple cell types in the tumor microenvironment. Intrinsically Cluster 5 also captures a positive relationship between CAIX, mTOR and pS6. In absence of clusters associated with resistance to treatment it was observed to be highly predictive of sensitivity to hormonal treatment.

Taken together, these observations exemplify how Kasumi allowed us to go from spatial measurements to potential biological insight, which can be applied to other related studies. We first detect persistent relationship patterns, identify the patterns associated with treatment response, contextualize them based on their frequency and spatial distribution, decompose these patterns in terms of the input features within and between cells to finally generate hypotheses about the underlying cellular mechanisms.

## Discussion

By establishing similarity based on common relationships across different spatial contexts instead of similarities in spatial profiles, Kasumi offers a representation that can be used for data exploration and hypothesis generation at the level of persistent neighborhoods and regarding their underlying mechanisms.

Spatially resolved data can improve patient stratification into treatable groups beyond clinical histology and beyond highly multiplexed data from dissociated tissues. Instead of characterizing samples based on standardized pre-selected clinical features, the Kasumi representation offers a meaningful and explainable association of localized relationship patterns to observable clinical outcomes. Kasumi representations can easily be adapted to explore tissue-specific common and differential patterns between conditions, even beyond association to clinical features.

On tasks of predicting disease progression and response to treatment, the performance of Kasumi shows the added value of considering spatially resolved data in contrast to cell-type distributions, significantly outperforming related methods. We further show that biologically relevant patterns are localized and that the local heterogeneity of tissues improves association to clinical features not only when considering cell-type level annotated samples, but also directly from measurements.

Kasumi offers an explainable framework to describe spatial data, highlighting both expected relationships and potential findings that can guide experimental follow-ups. We found that Kasumi clusters have more predictive power than simple local descriptions of cell type or marker composition. We achieved the best predictive performance on the prediction of immune checkpoint blockade response in the CTCL dataset. Of note, the most predictive clusters did not directly match the response criteria proposed in the original publication (distance between CD4+ cells and tumor cells, or between CD4+ cells and Tregs)[19]. These results illustrate how Kasumi allows users to look further than pairwise relations and identify more complex global and local patterns, exploring systematically cellular organization within and across samples. In agreement with the original publication, we do find that both the properties of the tumor and the lymphoid component of its immune micro-environment are needed to understand the response to immune checkpoint blockade. Altogether, our observations underscore the explainability of our approach and the potential of Kasumi for hypothesis generation.

Kasumi is flexible in terms of the input representation provided. We show that it achieves consistently high performance both in the scenario where input is provided at the level of cell types and at the level of marker abundances. Furthermore, we show that the relationship-based Kasumi representation outperforms composition-based representations considered by related approaches.

In the context of the more complex Kasumi output, when applied to marker-abundance input, we provide an approach to analyze Kasumi results that can also be applied in general to systematically derive biological insight from the generated outputs. First, we identified the clusters that are most relevant as predictors for a clinical task of interest. We next looked at the distribution of these clusters across samples to differentiate localized from globally present processes. As a result of the explainability of Kasumi's predictive model, we observed in more detail which specific intra- and intercellular patterns of relationships the clusters captured. We finally identified the compositional context captured by the clusters, ranging from cell-specific to relationships indicating more complex niches, as well as the scale of the processes they represent, ranging from sparse localized events to shared global interactions.

In this study, we show that Kasumi is applicable to high-resolution proteomics data. Our findings show that the availability of granular cell-type information can lead to better performance. However, if only high-level cell annotation is available then we expect that more information can be captured at the level of marker abundances. The same holds for measurements using technologies that are prone to contamination/dispersion of lineage markers to neighboring cells, preventing confident cell type identification. Since Kasumi is scalable, it can be run with both representations at input to compare the performance of the output on a relevant downstream task.

Of particular interest for further work is the application of Kasumi to different modalities. For higher dimensional data (e.g., transcriptomics) we recommend using well captured cell states that are

commonly assigned to segmented cells in samples measured with different high-resolution spatial omics technologies, such as merFISH or Xenium. Alternatively, to potentially generate more mechanistic insights, higher dimensional data can be transformed to lower dimensional representation by functional enrichment of the data. For example, by estimation of the activities of higher level biological processes, such as pathways[40], or capturing a subset of relevant molecules like receptor expression in the intraview and ligand expression in the paraview aiming at capturing spatially informed cell-cell communication patterns[41].

As means to an objective evaluation, we have focused on datasets with known clinical outcome and enough samples to pose a meaningful translational task. In particular, data with follow-up clinical data associated with them, but also data where the global non-spatial composition representation of distribution of cell types per sample cannot be reliably associated to the clinical outcome. Nevertheless, Kasumi can still complement the non-spatial analysis and generate insights. To demonstrate this and the scalability of Kasumi we applied it on a dataset of spatial molecular imaging (SMI, pre-commercial CosMx) comprised of a total of 300 fields of view with more than 700,000 cells taken from 7 untreated patients and 5 treated pancreatic ductal adenocarcinoma (PDAC) patients[42]. We show a summary of the results in Supplementary Fig. S11.

The application of Kasumi to low-resolution data requires working with aggregated gene expression or deconvoluted cell type data at input, which already represents a compositional neighborhood contrary to the relationship-based focus of Kasumi. While it is possible to run Kasumi in this context, the Kasumi analysis would then identify higher level neighborhoods. Nevertheless, while we considered view compositions capturing a single spatial context, Kasumi can be deployed with more complex compositions addressing different spatial and functional contexts, different omics technologies, and alternative non-compositional representations at input akin to recent applications of the global explainable relationship-based framework (MISTy)[43–48].

Another related direction for further work is exploring the integration of multiple modalities with Kasumi. Such modalities include not only different omics measurements but also image-based pathomics features. Thanks to the advancement of spatially resolved multiomics technologies[49–52], we can explore relationships across omics layers from the Kasumi representation to characterize the functional aspects of tissue patterns, which should lead to enhanced mechanistic insights.

Besides extending the layers, as datasets increase in number, we will be able to expand the complexity of the models. Higher-order (recursive) neighborhood analysis[53] is a step towards building models in the form of taxonomies of tissue structures. For this purpose, with Kasumi, we can incorporate both structural (cell-type level) and functional (marker level) aspects and derive higher-order relationships and representations. To explore the extent of preserved higher-order relationships with respect to different measures of similarity and towards establishing empirical taxonomies of tissue structure, Kasumi can also be combined with the existing related neighborhood analysis approaches.

In summary, Kasumi captures and accounts for multivariate and non-linear persistent spatially localized relationship patterns. It outperforms related approaches for neighborhood analysis, both at the level of cell types or without relying on cell-type labels. Kasumi is practically deployable even with a limited number of samples and can support early-stage translation efforts by targeted data exploration and hypothesis generation.

## Methods
### Data and preprocessing
The datasets used in this study were obtained from the papers in which they were originally presented.

The DCIS data was obtained from Risom et al.[15]. It consisted of 79 samples coming from 70 patients. From those, we selected all samples from 58 patients that were associated with either progressor status ($n = 14$) or nonprogressor status ($n = 44$). Samples from patients with normal or IBC-recurrence condition were excluded. The samples were segmented and annotated with 17 distinct sublineage labels based on the spatial profiles of the cells. The panel consisted of 38 antibody markers. For our marker abundance level analysis, we considered all of them without any further processing beyond the aggregation at the level of cells already available in the data.

The CTCL data was obtained from Phillips et al.[19]. It consisted of 69 samples coming from 14 patients. From those, we selected 29 pre-treatment samples coming from 14 patients associated with responder ($n = 14$) or nonresponder ($n = 15$) status. Samples taken post-treatment were excluded from the analysis. The samples were segmented and annotated with 21 distinct cluster labels associated with cell types, based on the spatial profiles of the cells. The panel consisted of 56 antibody markers. For our marker abundance level analysis, we considered all of them without any further processing beyond the aggregation at the level of cells already available in the data.

The BC data was obtained from Jackson et al.[33]. We considered only the Basel cohort due to the availability of the required metadata. It consisted of 376 samples coming from 285 patients. From these, we selected 85 samples coming from the tumor region of 82 patients with ER+PR+HER2- clinical subtype, treated with hormonal therapy and associated with sensitive ($n = 68$) or resistant ($n = 17$) response. To obtain a balanced dataset we further randomly subselected 15 samples from each group. The samples were segmented and annotated with 27 distinct cluster labels. The panel consisted of 33 antibody markers. For our marker abundance level analysis, we considered all of them without any further processing beyond the aggregation at the level of cells already available in the data.

The PDAC transcriptomics data (SMI, pre-commercial CosMx) was obtained from Shiau et al.[42]. From the total of 13 patients ($n = 6$ treated, $n = 7$ untreated), we removed samples from one treated patient that in addition to the capecitabine or 5-fluorouracil (CRT) treatment also received losartan treatment (CRTL). This resulted in selection of 300 fields of view, which we treated as independent samples. To represent each sample at input to Kasumi, we used the most granular cell type information for more than 700000 cells identified from the expression of a panel of 960 genes.

### Importance estimation
The importance of the estimated predictor-target relationships $M_{kj}$ by Kasumi and MISTy is based on the total reduction of the target ($k$) variance by the predictor ($j$) in the underlying target and view specific Random Forest model[30]. It is calculated as the view ($v$) contribution weighted standardized importance

$$M_{kj}^{(v)} = \frac{I_{kj}^{(v)} - I_k^{\overline{(v)}}}{\sigma_{I_k^v}^2}(1 - p_k^{(v)}),\tag{2}$$

where $I_k^{\overline{(v)}}$ is the mean, $\sigma_{I_k^v}$ the standard deviation of the estimated predictor importances for target $k$ and view $v$, and $p_k^{(v)}$ is the $p$-value of the contribution of view $v$ for target $k$ in the multi-view model shown in section 2.1.

### Kasumi importance signatures
The Kasumi importance signature for each window $k$ is represented as a vector generated by concatenating the estimated importances for each predictor-target pair from all views. The resulting vector is filtered by setting values to 0 for the predictor-target pair where the gain of variance explained for the target was estimated to be less than 1%. The gain of variance explained per target is a proxy of the amount of

information available for that target. If no relevant information was captured by the model then the estimated importances are also not relevant. The signature vector was additionally filtered such that it does not contain any values less than zero, i.e., does not contain information about interactions with less than the mean value across all importances.

## Representations and sensitivity analysis

For the predictive task, we summarized all representations in the same way, by the distribution of assigned identities per sample.

CSEA provides a neighborhood-based representation at the level of a whole sample via the estimated significance of interactions across all cell-type pairs. CCN, UTAG, BANKSY and CellCharter on the other hand provide labels at the level of single cells. In order to be applicable to the downstream tasks we represent each sample by the composition of assigned labels.

The global MISTy representation consists of the estimated importances of the relationships across all views for each sample. This is analogous to the first Kasumi representation if the whole sample has a single window. The relationship between the paraview parameter and the performance of the multi-view models was established in Tanevski et al.[27]. The paraview parameters were set in the same way as reported for Kasumi and kept same for all targets.

For the Kasumi analysis, we considered windows with a number of cells approximately equal to or larger than the number of variables in the intraview. We established a concave relationship between the selected window size and the performance of the Kasumi representation for the predictive tasks (Fig. 2b). Therefore, we selected the window size for each task to correspond to the size associated with the best performance.

The clustering of the first Kasumi representation depends on the choice of two parameters. First, the choice of a similarity cutoff for generating a graph representation of the signatures. In the graph representation, the edges between nodes with cosine similarity less than the threshold are removed. The second parameter is the resolution parameter of the Leiden community detection. We further explored the sensitivity of the Kasumi representations to the choice of these two parameters as measured by the performance on the predictive tasks by a grid search and selected the values that lead to the highest performance for each task (Supplementary Fig. S9).

## Parametrization of related approaches

The parameters of CSEA and CCN were set to the values reported as optimal in the original studies describing the method or its application on one of the datasets used in this study. The size of the neighborhood and the number of clusters is chosen to match the information given in the original data and method papers. In particular, 100 permutations for CSEA, a cell-centric neighborhood of 10 nearest cells, and a choice of 17, 10 and 6 clusters for the DCIS, CTCL and BC datasets respectively for CCN.

The size of the window and number of clusters for the Window Composition Clustering (WCC) was set to match the best performing values after running Kasumi to allow for direct comparison.

We ran BANKSY[23] both in cell typing an domain segmentation mode, by setting the lambda value to 0.2 and 0.8, respectively, as suggested for high resolution data. The value of $k_{geom}$ was set to 10, matching the setting for Kasumi and the other related approaches. For both modes, the resolution parameter for the Leiden clustering was fitted for best performance on each predictive task in the same way as for Kasumi. In Fig. 4 we report the overall best performance.

CellCharter[24] was run with one and three layers around each cell to capture a more local or broader tissue view. To determine the optimal number of clusters we performed 20 runs for each value of $k$ in the range of 2–15. We selected the range based on the maximum number of resulting number of Kasumi clusters across tasks. We selected and

report results for the most stable value for $k$ according to the similarity of repeated clustering measured by the Fowlkes-Mallows index, as suggested in the CellCharter manuscript.

For UTAG[25] we fit the resolution parameter for the Leiden clustering for the best performance on each predictive task in the same way as for Kasumi. We set the normalization to $L_1$ and a maximum distance of 20 pixels between pairs of cells.

## Signed model reliance

The relevance of each Kasumi cluster for the prediction task is estimated by the signed Model Reliance (sMR). Derived from permutation-based and model-agnostic Model Reliance[35], in the context of a linear (logistic) model sMR is defined as:

$$\text{sMR}(f, D, i) = \text{sgn}(\beta_i) \frac{L(f, D_{[p(i), \cdot]})}{L(f, D)}, \tag{3}$$

where $\beta_i$ is the estimated coefficient for the $i$-th variable for the linear (logistic) model f given unpermuted data D, $L(f, D) = 1 - \text{AUROC}(f, D)$ is the loss function, $p(i)$ is a permutation of the values of the $i$-th variable only. The AUROC estimate is macro averaged, by calculating it on concatenated predictions from a 10-fold cross-validation evaluation.

The model reliance[35] estimates the relevance of a variable by the change in the performance of the model under noise conditions. Here, the noise is introduced by permuting the values of one feature at a time and estimating the loss relative to the performance on the unpermuted dataset. The higher the loss, the more reliant the model is on that particular feature.

## Computational complexity

Let $m$ and $n$ be the length and width of a sample in given units of measurements (for example pixels, $\mu m$ or spots) in two dimensions. The total number of windows $z$ of size $s$ sliding with a fraction of overlap $o$ in both dimensions is then $z = ab$ where $a = \lceil \frac{m}{s(1-o)} \rceil$ is the number of windows per row and $b = \lceil \frac{n}{s(1-o)} \rceil$ is the number of windows per column. The $k$-th window then contains all units with coordinates $x \in [(k \bmod a)s(1-o), (k \bmod a + 1)s(1-o)]$ and $y \in [\lfloor k/a \rfloor s(1-o), (\lfloor k/a \rfloor + 1)s(1-o)]$.

The predictive model that is used by Kasumi is Random Forest which has a computational complexity $\mathcal{O}(tf^{\frac{1}{2}}p \log p)$, where $t$ is the number of trees in the ensemble, $f$ is the number of features (cell types or markers) and $p$ is the number of points in the dataset. For our analyzes we fixed the number of trees to 100. The selection of $\sqrt{f}$ number of features per split is a commonly recommended choice. For a single slide, given a window size equal to the size of the sample (comparable to a MISTy run), Kasumi constructs independent models for each feature in the intraview using all features from each view $v$ resulting in a complexity of $\mathcal{O}(vtf^{\frac{3}{2}}c \log c)$, where $c$ is the number of cells in the sample. Finally, given a total number $z$ of windows per sample, the computational complexity of Kasumi per sample is $\mathcal{O}(zvtf^{\frac{3}{2}}c_w \log c_w)$, where $c_w$ is the number of cells per window. The factor of added complexity due to the sliding window approach compared to running Kasumi on the whole slide is therefore $zc_w \log c_w / c \log c$.

In our experimental setup, running on a newer generation laptop with 8 processing cores, the time taken per Kasumi run on the samples from the studies analyzed in this paper with cell-type information was 8.9 (standard deviation 3.4) seconds per sample. Running Kasumi on a sample with marker abundance level information was 30.5 (standard deviation 24.9) seconds per sample. The differences in running time come from the larger number of markers compared to cell types and the different number of modeled views (one vs. two) in the two scenarios. Note that, as also stated in the Results section, in the case of one-hot encoded cell types, we don't model the intraview, as such a

prediction is trivial. More detailed information on the runtime per sample as function of the number of cells and features is shown in Supplementary Fig. S10.

Kasumi is highly parallelizable on different levels. Each sample from a dataset is run independently and can be run in parallel. For a single sample, each window can be modeled independently and in parallel. Finally, each view-specific model for every target in each window and sample can also be run in parallel.

### Reporting summary
Further information on research design is available in the Nature Portfolio Reporting Summary linked to this article.

## Data availability
The datasets used in this study have been made freely available as part of their original publications. The DCIS data was obtained by Risom et al.[15] and is publicly available from Mendeley at https://doi.org/10. 17632/d87vg86zd8.3. The CTCL data was obtained by Phillips et al.[19] and is publicly available as Source data from the original manuscript. The BC data was obtained from Jackson et al.[33] and is publicly available from Zenodo at https://doi.org/10.5281/zenodo.3518283. The PDAC data was obtained from Shiau et al.[42] and is publicly available from Mendeley at https://doi.org/10.17632/kx6b69n3cb.1. Source data are provided with this paper.

## Code availability
Kasumi is implemented as a modular open-source R package freely available from GitHub at https://www.github.com/jtanevski/kasumi under GNU General Public License v3.0. The code of the implementation of Kasumi that was used to analyze the data and produce the results is freely available from GitHub at https://www.github.com/saezlab/kasumi_bench under GNU General Public License v3.0. The specific version of the code associated with this publication together with streamlined examples of running Kasumi on the CTCL CODEX dataset and the PDAC SMI dataset and all result databases is archived in Zenodo[54] and is accessible at https://doi.org/10.5281/zenodo. 14891956 under GNU General Public License v3.0.

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

## Acknowledgements
We would like to thank Ricardo Omar Ramirez Flores, Philipp Sven Lars Schäfer, Robin Fallegger and Sebastian Lobentanzer for the helpful discussions. We acknowledge the financial support by the "Bruno und Helene Jöster Stiftung" (J.T.), the state funds approved by the State Parliament of Baden-Württemberg for the Innovation Campus Health + Life Science Alliance Heidelberg Mannheim (L.V.) and the German Federal Ministry of Education and Research through grant number BMBF 01ZZ2004 (D.S. and M.A.I.A).

## Author contributions
J.T.: Conceptualization, Methodology, Software, Validation, Formal analysis, Writing - Original Draft, Writing - Review & Editing. L.V.: Software, Formal analysis, Writing - Review & Editing. M.A.I.A.: Software, Visualization. D.S. Supervision. F.J.H.: Validation, Writing - Review & Editing, Supervision. J.S.R.: Supervision, Resources, Writing - Review & Editing, Funding acquisition.

## Funding

## Competing interests
J.S.R. reports in the last 3 years funding from GSK and Pfizer, and fees/honoraria from Travere Therapeutics, Stadapharm, Astex, Pfizer, Owkin, Moderna and Grunenthal. D.S. reports funding from Cellzome, a GSK company and received fees and honorariums from Immunai, Noetik, Alpenglow and Lunaphore. The remaining authors declare no competing interests.
