## [Transparent Peer Review file · Nature Communications]

Learning tissue representation by identification of persistent local patterns in spatial omics data

Corresponding Author: Dr Jovan Tanevski

Version 0:

Reviewer comments:

Reviewer #1

(Remarks to the Author)

In this manuscript, Tanevski et al. present the method Kasumi for identifying low-dimensional representations of spatial tissue samples that can be used for tasks such as predicting disease progression and treatment response from different samples. By considering three datasets, each of which has a number of samples from different patients, they are able to demonstrate that Kasumi outperforms alternative representations for such tasks. In particular, Kasumi uses an “explainable multi-view framework” building off of the authors’ previous method MISTy.

The manuscript seems to show that the Kasumi representations can be used for sample-based classification tasks such as disease progression and treatment response. However, it is held back by a lack of clear definition of what actually sets it apart from the prior method MISTy, given that it seems to use the same framework and even essentially the same name. The view-based framework requires the reader to keep track of many different terms (intraview, paraview, etc.) that are unclear and make it very difficult to understand what is going on, and the visualizations are similarly difficult to understand. While the explainability and interpretability of the representations are repeatedly emphasized in the manuscript, I find myself still confused as to what I am looking at in visualizations of the output.

Overall, the way in which the manuscript describes the method and results needs to be rewritten so that the reader is left with a clear understanding of what scientific findings have actually been made, and why the unique framework used is important to getting them. Specific points follow below:

Major comments:

1. Given the apparently high similarity of the Kasumi and MISTy frameworks, the authors must very clearly explain what aspects of the Kasumi method are actually different or new in order to establish the novelty of the manuscript.
2. The output representation of Kasumi needs to be much more explicitly defined in Figure 1/the relevant results section. Does the representation correspond to each sample, each spatial unit or each cluster? What is the size of representation? How does the visualization of the output be interpreted in terms of the original samples? What kind of biological problems can this method address?
3. Together with the last point, all terminology and description regarding “views” needs to be better explained. What are intraview, intrinsic view, two-view composition, broader tissue views etc.? All these terms need to be carefully defined and described in the main text, and better visualization of the key concepts in Fig. 1 is necessary.
4. The mechanism by which Kasumi uses feature importances to construct a representation is confusing. What kind of information does this type of feature importance representation provide? Why can the views not be used as features directly? It is not common to use feature importance for featurization in machine learning, and it is hard to intuitively understand what this representation means. More explanation needs to be given. “Filtering the set of all relationships to include only those with the highest importance results in a global relationship-oriented representation of the sample. For example, one can select only the predictor-target relationships one standard deviation above the mean importance per target and view. Simply put, a global relationship-oriented representation of a sample is the set of the most robust predictor-target relationships and their estimated importances.”
5. It seems like Fig. 2d is trying to show the essence of how Kasumi differentiates between responders and non-responders. Overall, the various boxes of different colors are kind of cryptic and don’t make the point of “explainable” – this visualization should be adjusted to reinforce how Kasumi is explaining the difference.
6. It seems “persistence” is a key component as given in the title, but it is not clear in what manner the representation actually

represents this persistence. What kind of property can this third layer of representation from persistence describe?

7. In Kasumi, features (i.e., gene expression) are aggregated into a coarse resolution, such as entire sample. The heterogeneity is omitted. How can local patterns be recognized from the representation for an entire sample?

8. Computational time is a big challenge for ensemble model (i.e., random forest) applying to large datasets. Especially, the random forest needs to be run permutatively over all “views”. The authors need to discuss the computational costs for each dataset.

Minor comments:

9. In Section 2.1, what are “all other variables (predictors)”?

10. In Section 2.1, too many formulas are given for the slide windows. Many of them are redundant and make it more difficult to understand the method.

11. The authors emphasized that their machine learning model is multivariate and nonlinear. But a majority of machine learning models are multivariate and nonlinear, and the random forest used in this work is a very common machine learning models.

12. In Section 2.2, why logistic regression? The sample sizes are very small in all tasks (<100). The classifier needs to be carefully studied for the underfit and overfit issues. Does the superior performance of Kasumi rely on the selection of model? What are dimensions of representation used for methods used for comparison?

13. In Section 2.3, the explainable phenomenon looks like the relationship between cell types and the diseases. How is this related to Kasumi if no information has been provided by Kasumi?

14. In Section 2.3, what is cluster-specific model? Cluster hasn't been discovered in previous sections.

15. In Section 2.3, what are M1 and M2?

16. It would be helpful if the section titles more clearly overviewed the relevant scientific findings.

17. Two Github links provided are unavailable.

(Remarks on code availability)

Reviewer #2

(Remarks to the Author)

Tanevski et al described a spatial clustering and tissue classification algorithm, called Kasumi. Domain clustering, disease detection, and patient stratification tools are extremely important for understanding tissue and disease biology, potentially improving clinical workflow. However, the manuscript is very hard to understand, with poorly described method. From the results presented in the figures, I am not convinced that Kasumi is an important advance. It seems to be just extending MISTy analysis to multiple windows followed by clustering. The authors also provided little comparison or benchmarking against existing literature and other clustering methods. For these reasons, I do not recommend publication without substantial revision. My questions for the authors (in no order of importance).

1. Fig 1 does little to help the reader understand the method, and may even lead to further confusion for the reader. Some suggestions:

a. Can you explain all the individual components in the Fig.1 schematic drawing? There are only broad uninformative titles (e.g. multi-view relationships) while the components within the figure (e.g. colours of cells, nodes and edges of relationship similarity graph) are unlabeled. The authors should clearly label each component or provide a legend.

b. It would be good to split the figure into sub-figures (a, b, c) for legibility and explain each sub-figure in detail in the corresponding figure caption.

c. What do the colors represent? Can the authors provide a description in the caption? The colours of the cells seem to correspond to cell types while colours of the windows on the right appear to be cluster labels, yet there are overlaps (e.g. purple and red). It would be helpful to use a different set of colours for different concepts or steps and provide a colour legend.

d. What do the tables or matrices represent, etc? Again, the columns/rows and the matrix itself are not labeled.

e. Could the authors provide a concrete example of what a view represents, perhaps as a subfigure?

2. To help readers understand the Kasumi method better, can you provide a flowchart of the method? Perhaps a detailed or full flowchart in the supplementary information and a simplified version in main figure 1? What are the inputs and outputs for each step?

3. How many ‘modes of operation’ does Kasumi has, and which mode is used for which figures/analyses?

4. How many parameters need to be defined by the users? How are the parameters tuned for each analysis/figure?

5. What exactly are the features used to calculate the Kasumi clusters?

6. Can the authors compare the clustering output when using the importance score versus simply using the average count/expression values? Perhaps compute the Adjusted Rand Index (or other index), and show spatially the differences between the clusters’ location when using each method?

7. How is the baseline defined? There seem to be at least two different definitions of baseline: 1) “the simple cell-type distributions per Sample” or 2) “sum of abundances per marker relative to the total abundance in the sample”.

8. Contributions of the different views: can the authors show how the multi-views framework impacts the clustering output and subsequent classification tasks? Perhaps also using ARI and spatial maps examples?

9. How is the persistence criteria defined? Is this a user parameter? This seemed to be a key step, but not well explained and demonstrated.

10. Can the authors provide the exact sample size versus the number of parameters (# of clusters) for the task of distinguishing responders from non-responders (Fig 3.) and tasks of predicting disease progression and response to treatment (Fig.4)? Can the authors show that they are not overfitting to the data? Maybe split into training and test sets?

11. In a supplementary figure, can the authors show all the cluster maps (spatial location of all the clusters) for all the tissues analyzed?

12. In addition to multiplexed protein data, can the authors demonstrate the use of Kasumi for transcriptomics data (e.g. Visium datasets)? The authors already showed some analysis of transcriptomics data for MISTy.

13. Can the authors compare the performance of Kasumi against other cutting-edge clustering algorithms like SpatialSort (Lee 2023), Cellcharter (Varrone 2023), BANKSY (Singhal 2024), UTAG (Kim 2022), and STELLAR (Brbić 2022)?

(Remarks on code availability)

Reviewer #3

(Remarks to the Author)

Tanevski et al. present Kasumi, a new method for identification of micro-environment in spatial omics data. The majority of methods in this area propose clustering patches from spatial-omics images to define distinct micro-environments. Their suggestion is to represent each patch by the multivariate cell type interaction in it. The multivariate cell type interaction is generated by using MistyR (Tanevski et al., 2022), an approach that utilizes feature importance of machine learning models in order to learn the relation between the cell types.

The idea of looking at the association between cell types / protein expression is very nice and extending Misty from entire FOVs to patches is an important development of the approach. Nevertheless, I have some reservations regarding the performance of the approach as detailed in the comments below. I will also note that the paper lacks clarity and details.

Major comments:

1. The paper lacks clarity and details in describing the algorithm. It was very difficult for me to understand the workflow, which is critical given that this manuscript showcases a new method. Figure 1 is very general and the current diagram could be used as is for showing almost any microenvironment detection method. To understand the method, I had to read the Misty manuscript, which is an unreasonable demand from the readers. I would like to see a much more detailed explanation of the method, including a detailed schematic and an illustrative example on actual data. Any technical details which are critical for understanding Kasumi should be presented and discussed, rather than just referring to the Misty manuscript.

2. It is my understanding that many cells are needed for training the ML models. If in Kasumi this model is now constructed for each window individually, then how are there enough statistics in e.g. 100 pixels and how is overfitting prevented? Does each window have enough statistics for a reliable model for all cell types / proteins?

3. The baseline employed in the paper, which is the global cell type composition per sample, differs from the common practice in the field. Typically, labs utilize the cell type composition of patches within images rather than the entire images, as done in this paper. I recommend including a comparison to this baseline, as it may reduce the disparity between Kasumi and the baseline and better reflect the results. This would also provide readers with valuable insights by comparing to the most widely adopted approach in the field.

4. In figure 2c,d the authors show images of tissues colored according to either cell types or Kasumi clusters. At least visually, there is a weak correspondence between the two. It seems that regions that look similar in terms of cell composition are often assigned to different clusters and vice versa. Can the authors explain these disparities? Can they delve deeper into the Kasumi clusters and interpret what drove them?

5. Related to the previous comment, the authors demonstrate better performance of their method compared to other approaches. However, their evaluation only focuses on accuracy for specific tasks. I suggest providing further comparison details. Are there new clusters that hold significance, or is the improved accuracy primarily due to more precise and accurate cluster assignments to cells?

6. The field of ME detection has exploded in recent years, but the authors compare Kasumi to older approaches. Many methods don't rely only on the protein expression. For example, many of the GNN methods. The GNN methods also have an un-supervised mode followed with fine tuning on a small set of labels. It's worth noting that there are several more recent approaches not addressed in the paper, including UTAG, CellCharter, CytoCommunity, and others. Some of these methods do not necessitate prior clustering or extensive labeled data and offer aspects of explainability. It would be beneficial to include comparisons with these approaches as well.

7. ME analysis is one stage in the pipeline of spatial transcriptomics/proteomics, Often the data is further analyzed for the functions of individual cells in the microenvironment, including looking at expression of specific programs in specific cell types, ligand-receptor interactions etc. In Kasumi, it seems that a cell isn't necessarily assigned to a unique ME. At least from figure 2 it seems that the sliding widow analysis may assign a cell to different, overlapping MEs. How do the authors suggest that such data could then be incorporated in downstream analyses?

8. Adding running time analysis (including the running time of Misty) would be beneficial for the readers.

Minor comments:

- In the introduction, the authors state certain flaws in current approaches, such as relying on the underlying clustering of cell types and post-hoc ME annotation. This is not entirely correct, as many methods can now incorporate the underlying expression values. Moreover, Kasumi similarly works on top of cell types and requires post-hoc cluster annotation. In fact, utilizing cell types doesn't seem like a problem because when the authors switch to using proteins in figure 4, their performance in predicting clinical parameters mostly drops.

- The authors might consider rewriting, simplifying and shortening the abstract for clarity.

- Regarding the equations in page 5:

o The analysis of the number of sliding windows in page 5 is trivial. It could be move to the methods or supplement, as it's not very interesting or relevant for the average reader.

o The equations explaining the representation of window k are not clear.

- o Consider separating the text and the equations to different rows.
- o Explain more clearly what each letter means.
- o There is an overload of letter definitions in few lines. Consider simplifying the equations and/or moving them to the methods.

(Remarks on code availability)

Reviewer #4

(Remarks to the Author)

(Remarks on code availability)

Version 1:

Reviewer comments:

Reviewer #1

(Remarks to the Author)

In the revision, the authors made extensive efforts to improve their work. The novelty of using niche/local neighbors (“view” by the authors) to generate representation of spatial data remains unclear, as similar approaches exist (e.g., MENDER doi.org/10.1038/s41467-023-44367-9). The overall significance of this work requires further demonstration.

Despite the relatively simple algorithm, its description is unnecessarily lengthy (spanning four pages for Section 2.1) and is written with imprecise machine learning and mathematical language along with very limited biological interpretation. Notably, the revised version has further reduced the number of figures from five to four, which further diminishes the clarity and depth of the analysis. Several questions raised in my initial review were only partially addressed. My comments are focused primarily on Section 2.1 and authors’ response letter. Significant works are needed for refinement and clarity of the manuscript.

1. (For comments 1-3 and 9-10) In Section 2.1, the description of “views” and the idea of “MISTy” seem to be clear. However, their description of “Kasumi” (their method), page 5-8, is still unclear. For example,
 - a. Why the two matrices in the right of Fig 1a have different type of column names in “predictor”?
 - b. The paragraph “What is persistence, can it be visualized, quantified? Similarly, what is “similarity”?”
 - c. (starting from first paragraph in page 7) The reduced/simplified version makes no improvement for clarity either in descriptions or mathematical formulation. For example, what is “meta-model”, “domain-specific functions”, “overlap o”, etc.. What are “i” and “j” for the importances matrices $M_{\{i,j\}^{\{v,k\}}}$? The authors describe index “i” as “target i” and “variable i in the intrinsic view” in the same equation, which is again confusing. The newly added paragraph “each sample is then represented by its compressed and relationship-oriented form of ...” can be deleted by simply describing the dimension of “M”.
 - d. (Methods) The authors moved content to Methods to describe computational complexity. But nothing is related to clarify their method.
 - e. In Section 2.1 and Methods, the authors need to find a way to present their overall ideas and the detailed algorithm in different levels.
 - f. The Random Forest utilizing “view” to predict “view” is a supervised model. Why the authors claim their method is unsupervised?
2. In Section 2.1, the authors now added definitions for “intraview” and “paraview” as two examples of “views”. Any other additional “views” were defined and used? Did the authors use the same set of “views” for all datasets tested in this work?
3. (Reply to comment 3): “All of the information needed to generate the explanations is already available in the Kasumi output and can be obtained by backtracking the relationship...” What are explanation of the model? How to visualize, quantify and interpret the “explanation” is unclear. What kind of biological insights can be explained by the model is unclear.
4. (Reply to Comment 6): It is still unknown what does the “persistence” mean in the manuscript. The reply from the authors reads like the “persistence” basically is the “clusters” of all windows over different samples. What kind of analysis and findings were discussed in this manuscript? Again, from my previous comment, what kind of property can this “persistence” provide?
5. (Section 2.2) “By defining a global, non-spatial baseline we aim to show that there are no significant differences...” What is the baseline method?
6. The “views” can be either cell types or given markers. But the authors only test the case using cell types as views. However, markers usually provide more information and do not require cell type annotation than “cell types”. Shall the authors present the results like the explanation using “markers” than “cell types”?
7. Why original Fig. 5 was removed?

(Remarks on code availability)

Reviewer #2

(Remarks to the Author)

I thank the authors for the detailed responses. I now have a better understanding of your work. The revised manuscript is much easier to read and I really appreciate the addition of potential biological insights (fig 4e). I have a few follow up questions:

- 1) For clarity and completeness, can the authors complete the flowchart on Fig. 1, indicating how the persistent clusters are being used for the logistic regression models predicting responders vs non-responders and progressors vs. non-progressors? What are the size and value of the input matrix
- 2) Why choose the value of 10 nearest neighbors? Have the authors tested other numbers? Should the number of neighbors be less than the sliding window size?
- 3) The estimated importance signature seems to serve as a kind of feature selection. Can the authors plot the importance score for one example dataset and show how many predictor-target relationships are one standard deviation above the mean importance per target and view?
- 4) The persistence criteria is a bit abstract and counter intuitive regarding how Kasumi can discover non-abundant relationships in the tissue. Can the authors show a plot from an actual data, how many and what are the clusters that did not pass the persistence criteria?
- 5) I really encourage the authors to demonstrate application in at least one spatial transcriptomics dataset. There are increasing number of high-resolution ST datasets with clinical phenotypes. For example, Shiau 2024 (DOI: 10.1038/s41588-024-01890-9) has SMI datasets for human PDAC tumors with (n = 6) or without (n = 7) neoadjuvant chemotherapy.
- 6) Discussion: do the authors have any recommendation or intuition as to which cell level representations of the samples, cell-type label or marker abundance level, is better? Should one always use the marker abundance mode.

(Remarks on code availability)

I was able to install mistyR and kasumi. However, I found it challenging to reproduce the analyses shown in the main figures based on https://github.com/saezlab/kasumi_bench. Can the authors provide some vignettes (similar to <https://saezlab.github.io/mistyR/articles/>) and a Zenodo archive with scripts to reproduce the analyses in the paper?

Reviewer #3

(Remarks to the Author)

Review for Tanevski et al.

The authors have addressed most of my comments. A few issues remain:

1. The authors added the window composition clustering as a baseline. However, both in the rebuttal and in the manuscript they refer to it as a novel idea (For example, "Our window composition clustering (WCC) approach" on page 15, or "such an approach can be useful to the community..." in the rebuttal). Performing window composition clustering is not novel. It is the baseline in the field. The authors should kindly modify the text in the paper to reflect this. Accordingly, they should rename the "Baseline" column in figures 2 and 4 to "Global composition", since global composition is not really the baseline. WCC is the baseline.
2. Running times: The authors added running times to the methods section. However, running times seem to be very large, averaging 8-30 seconds per cell. For a normally-sized dataset of 1M cells, running Kasumi will take 3 months to a year... The discussion doesn't include a limitations paragraph. The authors should add it and it should clearly indicate running times.

(Remarks on code availability)

Reviewer #4

(Remarks to the Author)

(Remarks on code availability)

Version 2:

Reviewer comments:

Reviewer #1

(Remarks to the Author)
No further comments.

(Remarks on code availability)

Reviewer #2

(Remarks to the Author)

The authors have answered all my questions. I also want to thank the authors for providing the example vignette. I have a minor comment about Supplementary Figure S10: It is a useful plot, but very hard to read. Consider changing from yellow to a darker color, try log scale for the y axis?, and please indicate which samples are being plotted. I have no other questions.

(Remarks on code availability)

Reviewer #3

(Remarks to the Author)

Review 3 for Tanevski et al.

1. I disagree. Almost every spatial proteomics paper that I know of used clustering in a small window to define microenvironments. Examples include (but are not limited to): Goltsev et al., Cell 2018 ; Schurch et al., Cell 2020 ; Hoch et al., Science Immunology 2021. I remain adamant in my previous request, not addressed by the authors:

- a. That the authors remove statements of novelty of the WCC approach.
- b. That the authors rename "Global baseline" to "Global composition". Like I said in the previous comments, this is not in any way the baseline in the field.

I believe that these requests are extremely reasonable and easy to implement.

2. The authors have addressed my comment.

(Remarks on code availability)

Reviewer #4

(Remarks to the Author)

(Remarks on code availability)

In the following, the reviewer's comments are shown in black while our responses are highlighted in blue. All text in quotation marks is taken from the revised manuscript. In the revised manuscript we also highlight the changes that resulted from addressing the comments in blue.

Reviewers' comments:

Reviewer #1 (Remarks to the Author):

In this manuscript, Tanevski et al. present the method Kasumi for identifying low-dimensional representations of spatial tissue samples that can be used for tasks such as predicting disease progression and treatment response from different samples. By considering three datasets, each of which has a number of samples from different patients, they are able to demonstrate that Kasumi outperforms alternative representations for such tasks. In particular, Kasumi uses an “explainable multi-view framework” building off of the authors’ previous method MISTy.

The manuscript seems to show that the Kasumi representations can be used for sample-based classification tasks such as disease progression and treatment response. However, it is held back by a lack of clear definition of what actually sets it apart from the prior method MISTy, given that it seems to use the same framework and even essentially the same name. The view-based framework requires the reader to keep track of many different terms (intraview, paraview, etc.) that are unclear and make it very difficult to understand what is going on, and the visualizations are similarly difficult to understand. While the explainability and interpretability of the representations are repeatedly emphasized in the manuscript, I find myself still confused as to what I am looking at in visualizations of the output.

Overall, the way in which the manuscript describes the method and results needs to be rewritten so that the reader is left with a clear understanding of what scientific findings have actually been made, and why the unique framework used is important to getting them.

We thank the reviewer for their comments. They raised important questions with regards to the distinction between MISTy and Kasumi as well as the clarity of the manuscript. We believe that by addressing these comments led to a much improved revised manuscript. Below we provide responses to the individual comments and outline the changes that were made.

Specific points follow below:

Major comments:

1. Given the apparently high similarity of the Kasumi and MISTy frameworks, the authors must very clearly explain what aspects of the Kasumi method are actually different or new in order to establish the novelty of the manuscript.

We acknowledge that our previous description might not have clearly explained the similarities and differences between MISTy and Kasumi. Our intention with Kasumi is to leverage the existing framework proposed by MISTy, extend it and instantiate it in a novel and distinct way, presenting a new approach to neighborhood analysis. This approach focuses on capturing the *local* heterogeneity of tissue structures and leveraging the representation for relevant downstream tasks that haven't been considered before within the MISTy framework due to the *global* nature of the relationships at the output of MISTy. Of note, MISTy is only a small part of the Kasumi workflow. The novelties introduced with Kasumi are (i) the focus on the local environment in contrast to the whole slide, (ii) the introduction of the concept of similarity between local neighborhood within and across samples that is possible as a result of the underlying relationship-based representation, (iii) the neighborhood-based representation of samples that is the result of clustering based on the similarity of relationships within a neighborhood in contrast to the similarity of cell-type or marker expression composition and (iv) the explainability and the ease of applicability of the resulting representation learned from different spatial omics technologies to relevant downstream tasks.

Based on these novel features, Kasumi outperforms related methods.

We now include a more detailed description of Kasumi in the introduction section with focus on the differences between MISTy and Kasumi, the novelties and the different types of tasks addressed by MISTy and Kasumi respectively. We updated Figure 1 and show explicitly where we use the MISTy framework to extract window-specific relationships. To further substantiate the differences to MISTy, we include it as part of the benchmarks too.

2. The output representation of Kasumi needs to be much more explicitly defined in Figure 1/the relevant results section. Does the representation correspond to each sample, each spatial unit or each cluster? What is the size of representation? How does the visualization of the output be interpreted in terms of the original samples? What kind of biological problems can this method address?

We updated Figure 1 and provide readers with a more detailed Kasumi workflow. We now show all data transformations and resulting representations. At the end of section 2.1 we now clarify that the representations are at the level of a window and describe them in more detail including the length of the representation per window. In particular:

“All Kasumi representations are at the level of a window (neighborhood) and are comparable across samples. Each sample is represented as a composition of

windows. Each window is represented first as a vector of importances of length equal to the total number of all underlying predictor-target relationships across views, second as a single cluster label, consistent across samples, and third, as the same cluster label if not filtered out based on the persistence criterion. Windows that are filtered out are simply removed from the representation and not considered further.”

In Figure 2 in c) and d) we show the representation of samples based on cell-type labels and Kasumi neighborhoods. Each window covers multiple cells and the label (color) assigned to the window is based on the clustering of the relationships extracted from that window. We updated the legend of Figure 2 to contain this information.

The representation used for the downstream tasks described in sections 2.2 - 2.4 is a summary of the window based representation of samples. Each sample is represented as the distribution of cluster labels, in the same way as the baseline approach is a distribution of cell types per sample and a distribution of neighborhoods for the other methods with the exception of MISTy. Since MISTy doesn't provide any neighborhood information, as representation we consider the importances of the relationships across all views for each sample. This is analogous to the first Kasumi representation if the whole sample has a single window. Our aim is to demonstrate the relevance of the local structures by exploring the range of neighborhood sizes (as shown in Figure 2b) with MISTy capturing the maximum possible neighborhood size, the sample itself. In the revised manuscript we now provide this information in section 2.2.

We developed the Kasumi representation to facilitate the analysis of local heterogeneity of tissues in health and disease. The use of Kasumi is initially envisioned to improve data exploration with focus on local instead of global relationships. The relationships coming from different spatial contexts point towards the robust biological processes underlying local patterns of organization. The Kasumi-based neighborhood analysis enables patient stratification by breaking down the heterogeneity of tissues. To this end we quantified and benchmarked its performance on tasks of associating relationship-based neighborhood composition to clinical outcomes. We outline these directions in the Discussion section.

3. Together with the last point, all terminology and description regarding “views” needs to be better explained. What are intraview, intrinsic view, two-view composition, broader tissue views etc.? All these terms need to be carefully defined and described in the main text, and better visualization of the key concepts in Fig. 1 is necessary.

In Figure 1 of the revised manuscript we now illustrate the different views, the information they capture and their composition that is the input to Kasumi. We

restructured the text, moved text from the Methods section to section 2.1 and 2.4, describing the views and their composition in more detail. We also added pointers in the text to the updated figure.

4. The mechanism by which Kasumi uses feature importances to construct a representation is confusing. What kind of information does this type of feature importance representation provide? Why can the views not be used as features directly? It is not common to use feature importance for featurization in machine learning, and it is hard to intuitively understand what this representation means. More explanation needs to be given. “Filtering the set of all relationships to include only those with the highest importance results in a global relationship-oriented representation of the sample. For example, one can select only the predictor-target relationships one standard deviation above the mean importance per target and view. Simply put, a global relationship-oriented representation of a sample is the set of the most robust predictor-target relationships and their estimated importances.”

We acknowledge that the terminology and concepts used can be difficult to follow, and we revised the manuscript and figures to try to make them clearer.

Some complexity is inherent to the fact that our approach proceeds in 2 steps: First, predictor-target relationships between the cellular features are modeled in different spatial contexts for each window in each sample. Second, we use the feature importances extracted from such models across sliding windows over all samples to serve as the basis of a quantitative representation of each window and sample. Feature importances are indeed not commonly used for featurization (feature construction), and we rather use them for feature selection, and the importances themselves are indicative of the strength of the relationship between the predictor and target. We now further clarify this point. For Random Forest models, this corresponds to the total reduction of variance achieved by each of the features when selected as splitting nodes in the constituent trees. Our framework simply uses this to identify meaningful relationships. Each feature and target in local models differ in scale, and making model coefficients comparable is a challenge. On the contrary, we can standardize the importances per target, such that they become invariant to scale and comparable across samples. Unlike feature transformation or feature construction, here the original variables and their meaning are preserved, so that we can further use them to explain the models. In the Methods section 4.2 we describe how we perform the standardization.

Since in the multiview modeling framework each view is also associated with a spatial context, the importances, and therefore the relationships, also get an interpretation of their spatial importance. In other works feature importances have been used successfully, for example in gene network reconstruction (e.g. Aibar et al. Nature Methods 2017), to estimate the edge weights in the network. Using feature importances in this manner has also been shown to improve the

performance and interpretability of larger neural network based approaches, where the importances are commonly estimated as part of the attention mechanisms although they remain explainable as long as they correlate with the feature importances estimated at the level of the model (Jain and Wallace arXiv 2019).

In the revised manuscript we rewrote the text and added more specific details about the importance estimation and aggregation that leads to the selection of the most robust predictor-target relationships.

5. It seems like Fig. 2d is trying to show the essence of how Kasumi differentiates between responders and non-responders. Overall, the various boxes of different colors are kind of cryptic and don't make the point of "explainable" – this visualization should be adjusted to reinforce how Kasumi is explaining the difference.

We agree that Fig 2.d was not clearly showing the explainability. We have updated the legend of Figure 2 to include more information about the correspondence between c) and d), and link to Figure 3, where the explainability is further elaborated. Each window covers multiple cells and the label (color) assigned to the window is based on the clustering of the relationships extracted from that window. The highlighted windows are taken from the results shown and further explained in Figure 3, identified as the most important neighborhoods to distinguish between responders and non-responders. We also updated the text of section 2.2 with information about the source of these samples and findings.

6. It seems "persistence" is a key component as given in the title, but it is not clear in what manner the representation actually represents this persistence. What kind of property can this third layer of representation from persistence describe?

With the introduction of the persistence criterion we address the trade-off between sample heterogeneity and the relevance of local structural patterns. Namely, in order for a neighborhood to be considered as meaningful and for further analysis and reasoning it should be present in more than one sample from the same tissue or condition. Multiple independent observations of the same pattern across samples adds to the relevance of the definition of the neighborhood, reduces the chances to include neighborhoods specific to a single patient rather than the tissue or condition, and it reduces potential sources of noise from the downstream analyses. In the revised manuscript we now include this information in the abstract and in section 2.1.

7. In Kasumi, features (i.e., gene expression) are aggregated into a coarse resolution, such as the entire sample. The heterogeneity is omitted. How can local patterns be recognized from the representation for an entire sample?

We hope that our response to comment 1 and the revised section 2.1 are now better emphasizing how Kasumi is accounting for heterogeneity, building on and refining MISTy, by extracting relationships present at the level of a window (smaller region of the sample). Therefore, the further analyses that are part of Kasumi - and not MISTy - are focused on the local patterns associated with the windows, which often vary significantly within samples.

In the revised manuscript we clarified the differences between the global MISTy analysis and the local nature of Kasumi among other distinguishing features. Furthermore, we included more information about the view construction that should resolve further unclarities both in the Results and in the Methods section. During the view construction in the case of the cell-type based analyses to construct the paraview, we aggregate information from the 10 nearest neighbors surrounding each cell. In the case of the marker-abundance based analysis we weigh the sum of marker abundances by a radial basis function of the distance between cells with a parameter set to 100px. By seeing if patterns are identified within windows, and are preserved in all or parts of a sample, we are directly encoding the sample heterogeneity.

8. Computational time is a big challenge for ensemble model (i.e., random forest) applying to large datasets. Especially, the random forest needs to be run permutatively over all “views”. The authors need to discuss the computational costs for each dataset.

We agree that computational costs are important, and we added a subsection on Computational complexity of Kasumi in the Methods section of the revised manuscript. We outline the computational complexity per sample as a function of the number of windows, views and number of features. We derived the added factor of complexity of our local sliding window approach in contrast to a global run on the whole sample.

We report that empirically “In our experimental setup, running on a newer generation laptop with 8 processing cores, the time taken per Kasumi run on the samples from the studies analyzed in this paper with cell-type information for each cell was $8.9(\pm 3.4)$ seconds. Running Kasumi on a sample with marker abundance level information was $30.5(\pm 24.9)$ seconds.”. We also added a supplementary figure showing the runtime per sample as a function of the number of cells and features per sample.

Note, also, that Kasumi is highly parallelizable on different levels. Each view-specific model for a target in each window and sample can be run in parallel. The underlying Random Forest models can also be parallelized, such that each tree in the ensemble is trained in parallel.

Minor comments:

9. In Section 2.1, what are “all other variables (predictors)”?

All other variables refer to the other features beyond the target variable that are available in the different views, i.e other cell-types in the neighborhood or abundances of all other markers within the cell or in the broader tissue structure as captured by the paraview. We added this information to section 2.1.

10. In Section 2.1, too many formulas are given for the slide windows. Many of them are redundant and make it more difficult to understand the method.

We simplified the description in Section 2.1, and moved the information and equations related to calculation of the number of windows from the main text to the new subsection Computational complexity in the Methods section.

11. The authors emphasized that their machine learning model is multivariate and nonlinear. But a majority of machine learning models are multivariate and nonlinear, and the random forest used in this work is a very common machine learning models.

As pointed out by the reviewer, multiple machine learning methods could be suitable to model the relationships between marker features in a multivariate and nonlinear manner. We chose Random Forest since, as the reviewer notes, it is very commonly used, shows good performance on a large variety of tasks, it is easily parallelizable and is explainable. We think that it is important to mention the nonlinearity of Random Forests as it affects the interpretation of the results, and at the same time we agree that this point does not need to be over-emphasized. We have rewritten the text to deemphasize this.

Of note, through the modular design of our framework, users can easily use other models if they find them to be more suitable to their data. As such, Kasumi is not limited to a single model choice. That said, we see this aspect of the choice of underlying machine learning model as largely independent from the core message of the current manuscript, namely that accounting for the distribution of spatial associations between features leads to more informative sample representations using the Kasumi framework.

12. In Section 2.2, why logistic regression? The sample sizes are very small in all tasks (<100). The classifier needs to be carefully studied for the underfit and overfit issues. Does the superior performance of Kasumi rely on the selection of model? What are dimensions of representation used for methods used for comparison?

We agree that the sample sizes are small, while at the same time realistic in terms of the currently publicly available omics data with coupled relevant follow up clinical information. We chose logistic regression not only for this reason, but also because we want to directly evaluate the relevant information captured by the Kasumi representation. We don't aim to perform further feature transformations and if the representation contains relevant biological information we shouldn't need a complex model to use it in a downstream relevant task.

In section 2.2 we note "The ability to use simple models based on a representation without further transformation suggests that the latter encodes relevant biological information and is easily interpretable."

To address potential under- or overfitting, the reported AUROC performance for Kasumi and all baselines and related methods is based on a macro ROC calculated on predictions made in a 10-fold cross validation evaluation scenario, as means to an objective evaluation. The macro averaging calculates the performance of the model across predictions pooled from all folds. The loss functions needed for calculating signed Model Reliance are also calculated on the predictions from a 10-fold cross validated evaluation, with and without perturbations. We clarified this in the revised text.

The dimension of the Kasumi representation is the number of unique clusters (neighborhoods) as the Kasumi representation used for this task is the distribution of clusters per sample. We now report the total number of persistent Kasumi clusters in the corresponding results subsections. We report the size of the datasets in section 2.3 and 4.1. The number of unique window clusters per dataset is consistently smaller than the number of samples, the number of cell-types or the number of markers.

13. In Section 2.3, the explainable phenomenon looks like the relationship between cell types and the diseases. How is this related to Kasumi if no information has been provided by Kasumi?

In section 2.3 we present explanations at two levels. First, the signed Model Reliance explains which Kasumi clusters are most relevant for the prediction task (here responders vs non-responders) given a description of samples based on the distribution of Kasumi neighborhoods (see also response to comment 2). At this level the neighborhoods (clusters) are learned by Kasumi. Furthermore, Kasumi offers an explanation of the relationships underlying each cluster. In section 2.3 we aggregated all windows across all samples corresponding to cluster 4, 13, 23 or 16 as clusters that were found as most relevant for the predictive task. Next we show the relationships with the highest importances as explanations of each cluster as well as the amount of variance explained per target that can be attributed to these

relationships. These importances also come from the Kasumi models that are learned per window (see also response to comment 4).

14. In Section 2.3, what is cluster-specific model? Cluster hasn't been discovered in previous sections.

To generate the second and third representation, Kasumi performs clustering of the importances for each window, i.e., the first representation, as explained now in more detail in section 2.1 and shown in the revised Figure 1. The distribution of Kasumi clusters per sample was used at input to the predictive tasks described in section 2.2 (see also response to comment 2).

15. In Section 2.3, what are M1 and M2?

The cell-type labels are taken from the original publication. The M1 and M2 labels are common descriptions of macrophages with two different polarizations. In the tumor microenvironment, the M1 macrophages have pro-inflammatory response to tumor cells, while the M2 macrophages are immunosuppressive and can be seen as pro-tumorigenic. In the text we note the anti-tumorigenesis role of M1>M2 and provide reference (33) to an overview article on this topic.

16. It would be helpful if the section titles more clearly overviewed the relevant scientific findings.

As the manuscript is primarily describing a method, we opted for subsection titles that highlight different features of the method. In the revised manuscript we changed the title of subsection 2.2 from "Kasumi preserves relevant biological information" to "Kasumi extracts robust patterns associated with clinical features" to align better with the other subsection titles. The subsection titles of the Result section now reflect the different aspects of the method that we present.

17. Two Github links provided are unavailable.

We apologize; we corrected the links. The issue was that the links were pointing to github.org instead of github.com which led to bad link forwarding.

Reviewer #2 (Remarks to the Author):

Tanevski et al described a spatial clustering and tissue classification algorithm, called Kasumi. Domain clustering, disease detection, and patient stratification tools are extremely important for understanding tissue and disease biology, potentially improving clinical workflow. However, the manuscript is very hard to understand, with poorly described method. From the results presented in the figures, I am not

convinced that Kasumi is an important advance. It seems to be just extending MISTy analysis to multiple windows followed by clustering. The authors also provided little comparison or benchmarking against existing literature and other clustering methods. For these reasons, I do not recommend publication without substantial revision. My questions for the authors (in no order of importance).

We thank the reviewer for the acknowledgement of the importance of the problem of neighborhood analysis and the relevance of the domain of application.

We would like to first make a clear distinction between our previous work and Kasumi. We acknowledge that our previous description might not have clearly addressed this. The differences are in terms of the general tasks solved by the approaches and the novelty of the introduction of relationship based representation learning of persistent neighborhoods; the definition of its supporting concepts and tasks, which to our knowledge haven't been addressed before, including the notion and properties of similarity in the new relationship-based representation space; the persistence criterion as control for false-positive detection of neighborhoods; and the evaluation framework demonstrating the use of the explainable features of Kasumi in an exploratory and translational settings. In the revised manuscript we added the following description in the Introduction section "MISTy is a general framework for extracting global relationships from spatial omics data. The output of MISTy is a set of robust relationships coming from different spatial contexts that are present across whole samples. The task commonly addressed with the MISTy framework is exploratory analysis and hypothesis generation. Kasumi extends the MISTy framework and instantiates it towards representation learning based on localized multiview relationships. With Kasumi we further address the task of neighborhood analysis by defining the concepts of similarity and persistence that are specific to the extended framework and the task of downstream learning by association of relationship-based representation of neighborhoods to clinical outcomes in a translational setting."

We extended the benchmark against three different clustering and neighborhood detection algorithms and introduced a local composition based clustering approach for direct comparison to the results from Kasumi. In the following we address the individual comments.

1. Fig 1 does little to help the reader understand the method, and may even lead to further confusion for the reader. Some suggestions:
 - a. Can you explain all the individual components in the Fig.1 schematic drawing? There are only broad uninformative titles (e.g. multi-view relationships) while the components within the figure (e.g. colours of cells, nodes and edges of relationship similarity graph) are unlabeled. The authors should clearly label each component or provide a legend.
 - b. It would be good to split the figure into sub-figures (a, b, c) for legibility and explain each sub-figure in detail in the corresponding figure caption.

- c. What do the colors represent? Can the authors provide a description in the caption? The colours of the cells seem to correspond to cell types while colours of the windows on the right appear to be cluster labels, yet there are overlaps (e.g. purple and red). It would be helpful to use a different set of colours for different concepts or steps and provide a colour legend.
- d. What do the tables or matrices represent, etc? Again, the columns/rows and the matrix itself are not labeled.
- e. Could the authors provide a concrete example of what a view represents, perhaps as a subfigure?

We agree that Figure 1 was too general and lacked the details to help the reader understand the method better. We redesigned Figure 1 taking into account the reviewer suggestions.

- a. We now explain the individual components in Figure 1, adding informative titles together with the appropriate legends.
- b. Figure 1 is now split into a, b and c subfigures. Each subfigure now explains a key aspect of the Kasumi workflow: a) The processing of sliding windows to their respective signatures, b) How the obtained signatures are used to construct a similarity graph and detect persistent communities, and c) The handling of samples in downstream tasks.

This is complemented with an extended caption of Figure 1:

“a) Kasumi takes as input a number of spatially resolved measurements from tissue samples, where each spatial location is assigned a cell type (alternatively a vector of abundances of measured markers (Supplementary Figure S1). Each sample is decomposed into tissue patches by sliding a window of fixed size with or without overlap. From each window, Kasumi extracts relationships coming from different spatial contexts by estimating the importance of each measured variable as a predictor of each target variable from a multi-view multivariate non-linear predictive model. b) The relationships per window across all samples are used to construct a relationship similarity graph. The windows are clustered by graph community detection followed by cluster removal based on a persistence criterion. The output of Kasumi are compressed explainable relationship-based representations of the samples preserving relevant biological signals. c) For downstream tasks each sample is represented by the distribution of persistent Kasumi clusters, capturing its local pattern composition. ”

- c. We now use a set of different colors to differentiate specific concepts in the figure while adding clearer color legends and descriptions.
- d. The matrices represent the window importance signatures. In the updated figure we updated the representations and added the corresponding labels.
- e. In the updated figure we provide concrete examples for the representation of intraview and paraview.

2. To help readers understand the Kasumi method better, can you provide a flowchart of the method? Perhaps a detailed or full flowchart in the supplementary information and a simplified version in main figure 1? What are the inputs and outputs for each step?

We redesigned Figure 1 to depict the workflow of Kasumi and the individual steps in more detail. The input to Kasumi is now shown in subfigure a) as a collection of samples with single-cell resolution. We then show the construction of the view composition and give an example of the matrix representation of the views for a single window. The example in Figure 1 is focused on a cell-type-based scenario and we further added Supplementary Figure 1 to show the differences in the input representation for a marker-based scenario. We next show explicitly the three different representations and the transformations needed to obtain them starting from the window signatures. The output of Kasumi and the representation of the samples for the downstream tasks are now shown in subfigure c).

3. How many 'modes of operation' does Kasumi has, and which mode is used for which figures/analyses?

There is a single Kasumi workflow which we now present in more detail in Figure 1 and the corresponding revised text of the Results section. The differences between the two groups of results presented in subsections 2.2 and 2.4 come as a result of using two different types of initial information per cell and sample. Namely in sections 2.2 and 2.3 we show results of running Kasumi on cell-type level information at input, while in section 2.4 we show results of running Kasumi on marker abundance level information at input. Both of these initial representations are commonly used in practice and Kasumi is able to learn meaningful neighborhood level representations from both.

In the revised manuscript we include more details regarding the construction of the view compositions used as input to Kasumi given the two different initial sample representations. We also added a paragraph at the end of section 2.1 stating the different inputs to the analyses that follow:

“We learn Kasumi representations of samples coming from three different cohorts of oncological patients measured respectively with three different spatial proteomics technologies. We learn neighborhood level sample representations starting from two initial cell level representations of the samples -- cell-type label and marker abundance level information for each cell in each sample with each of the three cohorts. To evaluate the proposed approach we compare the performance of the learned representation to several baselines and related approaches on the task of patient stratification. We measure the performance given a ground truth of follow up observations of disease progression and response to treatment.”

4. How many parameters need to be defined by the users? How are the parameters tuned for each analysis/figure?

For each sample the input parameters to the Kasumi representation learning are: (i) a view composition, (ii) window size and optionally (iii) a percentage of overlap and (iv) minimum number of data points per window.

In the revised manuscript we described in more detail the construction of the view composition for the two scenarios. In the two scenarios we explored different window sizes ranging from 100px up to the whole sample (MISTy global). The results shown per dataset are using the window size that maximizes the predictive performance of Kasumi. We describe this procedure in section 2.2. For the second scenario the window sizes remain the same to enable direct comparison between scenarios. The percentage of window overlap is 50% by default, trading off computational time with sample coverage. We discuss the effect of the total number of windows per sample on the computational complexity in the new subsection 4.6. To avoid overfitting due to the large p small n problem, we set the minimum number of points to window to the number of features in the intraview. We now include more details regarding these optional parameters and their default values in section 2.1.

The parameters of the clustering are similarity cutoff and the clustering resolution. We select the values of these parameters based on the performance of the resulting representation on the downstream tasks. We performed an analysis of the sensitivity of Kasumi to the selection of these parameters too (Section 4.4 and Supplementary Figure).

We now also include complementary information on the selection and tuning of the parameters of the related approaches, whose results we show in the main figures in the new section 4.5.

5. What exactly are the features used to calculate the Kasumi clusters?

We rewrote the paragraph describing the clustering approach in section 2.1. “.. we define and cluster a graph of similarities of all windows across all samples. Each node in the graph represents a window described by the Kasumi importance signature, i.e., the set of all estimated predictor-target relationships from that window with importance larger than zero (subsection Kasumi importance signatures). The nodes in the graph are connected by edges with weights equal to the cosine similarity between the node representations. The edges of the fully connected graph are filtered based on a similarity cutoff. The Kasumi clusters are then determined by Leiden community detection (see Predictive tasks and sensitivity analysis).”

6. Can the authors compare the clustering output when using the importance score versus simply using the average count/expression values? Perhaps compute the Adjusted Rand Index (or other index), and show spatially the differences between the clusters' location when using each method?

For the revised manuscript we implemented a new Window Composition Clustering (WCC) approach. In particular – “in order to directly estimate the value of the local relationship-based representation of Kasumi relative to a composition based neighborhood representation, we implemented a clustering of composition of sliding windows across samples. Instead of representing each Kasumi window by the estimated relationships, we represent it by the normalized vector of composition of cell types within that window. The resulting representation is comparable across windows and samples. As for Kasumi, we first remove the windows that contain less cells than the total number of cell types. We next cluster the windows and represent each sample by the frequency of its window clusters in the same way as for Kasumi. To ensure direct comparison to Kasumi, we set the window size and the number of clusters to match the best performing Kasumi run. Any performance gain of Kasumi in addition to the Window Composition Clustering (WCC) can therefore be attributed to the relationship-based representation capturing relevant interactions beyond composition.”.

All related methods produce cluster labels at the level of a cell, while Kasumi produces cluster labels at the level of a patch. Therefore the similarity or differences of the clustering results cannot be directly beyond their performance on downstream tasks. However, In addition to performance comparison WCC allows for the direct comparison of Kasumi and WCC clustering at the level of individual window across all samples.

As suggested by the reviewer, we now provide the distribution of Adjusted Rand Index (ARI) across all samples for each task in the two scenarios in the Supplementary Material and report the mean ARI in the main text. In the supplementary material of the revised manuscript we provide cluster maps for all samples from the datasets we are focusing on in more detail.

7. How is the baseline defined? There seem to be at least two different definitions of baseline: 1) “the simple cell-type distributions per Sample” or 2) “sum of abundances per marker relative to the total abundance in the sample”.

For the two different scenarios we include two global baselines due to the differences in the input representation. For the scenario where we run Kasumi on cell-type level information at input (Section 2.2. and 2.3) the baseline is the distribution of cell types per sample. For the scenario where we run Kasumi on marker abundance level information at input (Section 2.4) the baseline is the representation of the sample as a vector of the sum of abundances per sample. In both cases the representation is normalized such that the values per sample sum

up to 1. Therefore there is a noticeable difference in performance between the two baselines. We now relabeled the Baseline columns in figures 2 and 4 to Baseline (ct) and Baseline (marker) respectively, to highlight the difference between them.

In both cases the purpose of establishing a global, non-spatial baseline is to show that there are no significant differences between the groups of patients (responders vs non-responders, progressors vs. non-progressors) when the spatial context of the data is not taken into account. Consequently the improvement of performance, given the different neighborhood analysis approaches is due to considering the spatial organization of the cell types and the marker abundances, as well as the relationships between them within the different spatial contexts. We now include this information in the text of sections 2.2 and 2.4.

In the revised manuscript in both scenarios we now include additional local distribution based baselines, where for local representation of the sample we consider the cell type distribution or the sum of abundances per marker within windows (patches) of the samples that match locations of the Kasumi windows. With these baselines we aim to show the increase of performance on the downstream tasks that can be attributed to the relationship-based representation.

8. Contributions of the different views: can the authors show how the multi-views framework impacts the clustering output and subsequent classification tasks? Perhaps also using ARI and spatial maps examples?

In the cell-type based scenario the interview we do not model the interview since predicting cell-type identities from one-hot encoded representation is not informative. Therefore the reported performance in terms of variance explained per target cell-type comes from the paraview. In the revised manuscript, we now include Supplementary Figure S7 for the marker-abundance base scenario. This supplementary figure shows the impact on variance explained per target marker of adding the paraview in addition to the intraview. We further show the relative contribution of the views for explaining the variance of the selected targets for the most relevant clusters. We extended the text in Section 2.4, relating the results in Figure 4e with the results shown in the supplementary figure. Additionally, the extended reporting of the results in Section 2.4 is focused on explaining and comparing the relationships coming from the different spatial contexts in more detail. We further contrast our multi-view approach with a (single-view) composition based approach by the introduction of the Window Composition Clustering (WCC), the comparison of its performance, and the comparison of the results of the clustering (ARI).

9. How is the persistence criteria defined? Is this a user parameter? This seemed to be a key step, but not well explained and demonstrated.

Indeed the persistence criterion is one of the key concepts we introduce within Kasumi, and we acknowledge that it was not clearly introduced in the first version of the manuscript. We introduce the persistence criterion to address the trade-off between sample heterogeneity and the relevance of local structural patterns. We now define persistence already in the abstract and extend the description of the persistence criterion in section 2.1 as follows

“Local patterns are meaningful for reasoning in terms of knowledge discovery and application to downstream tasks if they are persistently present across samples. The multiple independent observations of the same pattern within a subset of tissues in or across conditions adds to the relevance of the neighborhood to that tissue or condition. It also reduces the chances to include neighborhoods specific to a single sample that add noise and bias to downstream analyses. To this end we retain clusters that are present in at least 10% or at least 5 samples.”

10. Can the authors provide the exact sample size versus the number of parameters (# of clusters) for the task of distinguishing responders from non-responders (Fig 3.) and tasks of predicting disease progression and response to treatment (Fig.4)? Can the authors show that they are not overfitting to the data? Maybe split into training and test sets?

In the revised manuscript we provide information for the size of the Kasumi representation for all tasks. For the cell-type based scenario “For the optimal window choice the learned Kasumi representation consisted of 33, 16 and 9 unique persistent window clusters for the DCIS, CTCL and BC datasets respectively.”. For the marker-abundance based scenario “For the optimal window choice the learned Kasumi representation consisted of 4, 12 and 15 unique persistent window clusters for the DCIS, CTCL and BC datasets respectively.” The corresponding sample sizes are 58, 29 and 30. We report the size of the datasets in section 2.3 and 4.1. The number of unique window clusters per dataset is consistently smaller than the number of samples. The resulting Kasumi representation per sample is the vector of distribution of window clusters for that sample.

The reported AUROC performance for Kasumi and all baselines and related methods is based on a macro ROC calculated on the concatenated predictions made in a 10-fold cross validation evaluation scenario, as means to an objective evaluation. The loss functions needed for calculating signed Model Reliance are also calculated on the predictions from a 10-fold cross validated evaluation, with and without perturbations. We clarified this in the revised text.

11. In a supplementary figure, can the authors show all the cluster maps (spatial location of all the clusters) for all the tissues analyzed?

In the supplementary material of the revised manuscript we provide cluster maps for all samples from the datasets we are focusing on in more detail. In particular, for the CTCL dataset we show the spatial locations of all cell types, the cell-type-composition-based window clusters and Kasumi clusters for all considered samples. We provide separate figures for the responder and non-responder samples. For the BC dataset we show the BANKSY clusters per cell (best performing related method on this task), marker-abundance-composition-based window clusters (WCC) and Kasumi clusters for all considered samples. We provide separate figures for the treatment-sensitive and resistant samples.

12. In addition to multiplexed protein data, can the authors demonstrate the use of Kasumi for transcriptomics data (e.g. Visium datasets)? The authors already showed some analysis of transcriptomics data for MISTy.

We appreciate the reviewer's suggestion. It was a direction that we also wanted to explore in the paper. As means to an objective evaluation, we have focused on datasets where we have clinical outcome data and enough samples to demonstrate its applicability to meaningful translational tasks. Unfortunately, we weren't able to find spatial transcriptomics datasets that fit the current evaluation framework. In particular, publicly available high-resolution spatial transcriptomics datasets either do not have follow up clinical data associated with them, have a very limited number (1-2) of samples per condition, or compare control to condition samples, which can be predicted without the need for spatial resolution or neighborhood analysis. The same issues affect the publicly available low-resolution datasets (ex. Visium). In addition, the analysis of low-resolution data would also be based on compositional data. Each spot is inherently represented by the composition of the gene expression of multiple cell types or requires deconvolution based on reference data, resulting again in estimated compositions of cell types, which limits the relevance of spatial dependencies between spots and is therefore not a use case in which we would recommend Kasumi. We acknowledge these current limitations of our work in the Discussion section and expanded the text to include more details and guidelines for future work.

"Of particular interest for further work is the application of Kasumi to different modalities. In this study, we show that Kasumi is applicable to high-resolution proteomics data. Furthermore, its best performance on downstream translational tasks is achieved when taking cell-type labels as input representation of each sample in the dataset. In principle, such application of Kasumi is not limited to proteomics, as cell type or even cell state level information is commonly assigned to segmented cells in samples measured with different high-resolution spatial omics

technologies, such as merFISH or Xenium. To this end, we don't anticipate any limitation to the use of Kasumi to analyze such data. The application of Kasumi to low-resolution data requires working with aggregated gene expression or deconvoluted cell type data at input, which already represents a compositional neighborhood contrary to the relationship-based focus of Kasumi. While it is possible to run Kasumi in this context, the Kasumi analysis would then identify higher level neighborhoods. Nevertheless, while we considered view compositions capturing a single spatial context, Kasumi can be deployed with more complex compositions addressing different spatial and functional contexts, different omics technologies, and alternative non-compositional representations at input akin recent applications of the global explainable relationship-based framework (MISTy).”

13. Can the authors compare the performance of Kasumi against other cutting-edge clustering algorithms like SpatialSort (Lee 2023), CellCharter (Varrone 2023), BANKSY (Singhal 2024), UTAG (Kim 2022), and STELLAR (Brbić 2022)?

In the revised manuscript for the marker-abundance based scenario we now compare the performance of Kasumi to BANKSY, CellCharter and UTAG. We selected recently published methods based on the suggestion of the reviewers, but also according to their reported performance in comparison to other related approaches. We optimized the parameters of all related approaches based on the performance on the downstream tasks (Section 4.5 Parametrization of related approaches). Kasumi outperforms CellCharter on two out of the three tasks and achieves comparable performance on the third one (AUROC CellCharter 0.79 vs. Kasumi 0.78) Kasumi outperforms BANKSY on all tasks. BANKSY shows comparable performance to Kasumi on one task (AUROC BANKSY 0.74 vs. Kasumi 0.77). Kasumi shows consistently high performance across all tasks.

Reviewer #3 (Remarks to the Author):

Tanevski et al. present Kasumi, a new method for identification of micro-environment in spatial omics data. The majority of methods in this area propose clustering patches from spatial-omics images to define distinct micro-environments. Their suggestion is to represent each patch by the multivariate cell type interaction in it. The multivariate cell type interaction is generated by using MistyR (Tanevski et al., 2022), an approach that utilizes feature importance of machine learning models in order to learn the relation between the cell types. The idea of looking at the association between cell types / protein expression is very nice and extending Misty from entire FOVs to patches is an important development of the approach. Nevertheless, I have some reservations regarding

the performance of the approach as detailed in the comments below. I will also note that the paper lacks clarity and details.

We would like to thank the reviewer for recognizing the importance of the task of neighborhood analysis and the distinctiveness of the relationship-based approach introduced by Kasumi. Below we provide responses to the individual comments.

Major comments:

1. The paper lacks clarity and details in describing the algorithm. It was very difficult for me to understand the workflow, which is critical given that this manuscript showcases a new method. Figure 1 is very general and the current diagram could be used as is for showing almost any microenvironment detection method. To understand the method, I had to read the Misty manuscript, which is an unreasonable demand from the readers. I would like to see a much more detailed explanation of the method, including a detailed schematic and an illustrative example on actual data. Any technical details which are critical for understanding Kasumi should be presented and discussed, rather than just referring to the Misty manuscript.

We agree with the reviewer that the original Figure 1 was too general and the method and the workflow require more details in the first figure and in the corresponding text. We have now significantly improved Figure 1 and provide a higher level of detail including the results of the data transformation at every step of the workflow. We added information on the method, parameters, decisions and tasks in the text.

2. It is my understanding that many cells are needed for training the ML models. If in Kasumi this model is now constructed for each window individually, then how are there enough statistics in e.g. 100 pixels and how is overfitting prevented? Does each window have enough statistics for a reliable model for all cell types / proteins?

To avoid overfitting due to the large p small n problem, we set the minimum number of points to window to the number of features in the intraview. For each window, models are trained only for those cell types that are present in that window. The persistence criterion additionally ensures that the resulting clusters are not sample specific, but persistent within the condition or cohort. We introduce the persistence criterion to address the trade-off between sample heterogeneity and the relevance of local structural patterns. We now define persistence already in the abstract and extend the description of the persistence criterion in section 2.1.

The reported AUROC performance for Kasumi and all baselines and related methods is based on a macro ROC calculated on concatenated predictions made in a 10-fold cross validation evaluation scenario, as means to an objective evaluation. The loss functions needed for calculating signed Model Reliance are also

calculated on the predictions from a 10-fold cross validated evaluation, with and without perturbations. We clarified this in the Results and Method sections of the revised text.

3. The baseline employed in the paper, which is the global cell type composition per sample, differs from the common practice in the field. Typically, labs utilize the cell type composition of patches within images rather than the entire images, as done in this paper. I recommend including a comparison to this baseline, as it may reduce the disparity between Kasumi and the baseline and better reflect the results. This would also provide readers with valuable insights by comparing to the most widely adopted approach in the field.

We would like to thank the reviewer for their great suggestion. We believe that by addressing this comment we were able to gain valuable insight into Kasumi and the added value of the proposed relationship-based approach by direct comparison to a composition-based window clustering approach. Different classes of approaches to analysis of spatial omics data focus on the analysis of specific anatomical regions, tissue layers or regions of interest. In the case of unsegmented high-resolution spatial-omics data, it is also common to adopt a binning approach, most frequently followed by cell-type calling or deconvolution to identify the distribution of cell types within the bin. To the best of our knowledge no other method for neighborhood analysis for highly multiplexed spatial omics data performs analysis at the level of regularly shaped and overlapping tissue patches containing multiple segmented cells across samples.

We however agree with the reviewer that a comparison with composition-based clustering of tissue patches adds a new and valuable composition-relationship aspect to the analysis in addition to the existing global-local axis. To address this we implemented a new Window Composition Clustering (WCC) approach.

In particular, “in order to directly estimate the value of the local relationship-based representation of Kasumi relative to a composition based neighborhood representation, we implemented a clustering of composition of sliding windows across samples. Instead of representing each Kasumi window by the estimated relationships, we represent it by the normalized vector of composition of cell types within that window. The resulting representation is comparable across windows and samples. As for Kasumi, we first remove the windows that contain less cells than the total number of cell types. We next cluster the windows and represent each sample by the frequency of its window clusters in the same way as for Kasumi. To ensure direct comparison to Kasumi, we set the window size and the number of clusters to match the best performing Kasumi run. Any performance gain of Kasumi in addition to the Window Composition Clustering (WCC) can therefore be attributed to the relationship-based representation capturing relevant interactions beyond composition.”.

In the results sections 2.2 and 2.4 we now report the performance of WCC and compare it with the performance of Kasumi. This analysis enabled us to pinpoint that the reduction in performance between running Kasumi at the level of cell-type labels and at the level of marker abundances on some datasets is due to Kasumi capturing composition instead of relationships. Running KWC with the same optimal window size and number of clusters as Kasumi enables a direct comparison of the clustering, which we report on in the Results section and complement in Supplementary Figures S2-S5 by visualizing the Kasumi and WCC clusters on all samples. In the revised manuscript we also include a Supplementary Figure S6 depicting the distribution of the Adjusted Rand Index across samples per input and datasets.

We believe that, as the reviewer points out, such an approach would not only add value to the manuscript but also can be useful to the community as an alternative to other neighborhood analysis approaches. We therefore made WCC part of the Kasumi package as a compositional alternative to our original relationship-based approach. The WCC implementation follows the same Kasumi pipeline, replacing the first Kasumi representation based on estimated importances of predictor-target relationships with the cell-type or marker abundance composition of each window.

4. In figure 2c,d the authors show images of tissues colored according to either cell types or Kasumi clusters. At least visually, there is a weak correspondence between the two. It seems that regions that look similar in terms of cell composition are often assigned to different clusters and vice versa. Can the authors explain these disparities? Can they delve deeper into the Kasumi clusters and interpret what drove them?

To further explore what drives the Kasumi clusters and demonstrate interpretability, we extended the analysis of the results and added more details especially for the more complex Kasumi output, when applied to a marker-abundance input in section 2.4. We start from the clusters that were identified to be most important for the translational task. Next, we looked at the distribution of the clusters across samples and made a note of their local or global nature. We then focused on the specific intra and intercellular patterns of relationships that the most relevant clusters captured and identified the compositional context captured by the clusters, ranging from cell-specific to relationships indicating more complex niches, as well as the scale of the processes they represent, ranging from sparse localized events to shared global interactions.

We thank the reviewer for these comments that led us to not only characterize a multicellular pattern underlying response to hormone treatment in breast cancer, but more generally demonstrate how to interpret Kasumi results. This is presented in the revised section 2.4, as well as in the added supplementary figures 3 and 4.

5. Related to the previous comment, the authors demonstrate better performance of their method compared to other approaches. However, their evaluation only focuses on accuracy for specific tasks. I suggest providing further comparison details. Are there new clusters that hold significance, or is the improved accuracy primarily due to more precise and accurate cluster assignments to cells?

We agree that the manuscript would be strengthened by further comparison of how Kasumi led to better prediction, and updated the manuscript accordingly. In particular in our application of Kasumi to marker intensities in the breast cancer data (section 2.4), we identified a pattern that was undetected by other methods. This is essentially covered by the following additions:

Kasumi was able to identify a 3-way pattern involving the immune, stromal and tumoral compartments as well as their relative spatial organization. By comparison, BANKSY captured a pattern involving GATA3 positive cancer cells (cluster 1, Supplementary Figure S5a, but it is not apparent from its output how they are organized with respect to other cell classes.

" [...] Our observation suggests that this regulation is not present in the whole tumor sample but acting locally within tumors with immune deserts. GATA3 expression is typically a marker of good prognosis. By its association with ER, it can be indicative of a response to hormonal treatments. However, this is not the case for the five samples in the cohort that contain cluster 37. The high fibronectin intensity relates to a dense stromal compartment that may prevent immune infiltration and the availability of ER. In such cases, the hormone treatment can potentially benefit from coupling with an intervention on the tumor-immune microenvironment."

The signed Model Reliance values are also a way to explore what would happen without a given cluster, hence to quantify their significance for any chosen task. We show the relevance of each cluster of windows in Figure 3 and 4. To clarify, we do not assign clusters to individual cells, but rather to the whole tissue patches (windows). The better performance was obtained, at least in this example, thanks to the ability to identify localized multicellular patterns and to dissect and differentiate the intrinsic cellular properties from the contribution of the cellular patterns of organization at a higher level.

6. The field of ME detection has exploded in recent years, but the authors compare Kasumi to older approaches. Many methods don't rely only on the protein expression. For example, many of the GNN methods. The GNN methods also have an un-supervised mode followed with fine tuning on a small set of labels. It's worth noting that there are several more recent approaches not addressed in the paper, including UTAG, CellCharter, CytoCommunity, and others. Some of these methods do not necessitate prior clustering or extensive labeled data and offer aspects of

explainability. It would be beneficial to include comparisons with these approaches as well.

We agree with the reviewer. In the revised manuscript for the marker-abundance based scenario we now compare the performance of Kasumi to BANKSY, CellCharter and UTAG. We selected recently published methods based on the suggestion of the reviewers, but also according to their reported performance in comparison to other related approaches. We optimized the parameters of all related approaches based on the performance on the downstream tasks (Section 4.5 Parametrization of related approaches). Kasumi outperforms CellCharter on two out of the three tasks and achieves comparable performance on the third one (AUROC CellCharter 0.79 vs. Kasumi 0.78) Kasumi outperforms BANKSY on all tasks. BANKSY shows comparable performance to Kasumi on one task (AUROC BANKSY 0.74 vs. Kasumi 0.77). Kasumi shows consistently high performance across all tasks.

7. ME analysis is one stage in the pipeline of spatial transcriptomics/proteomics, Often the data is further analyzed for the functions of individual cells in the microenvironment, including looking at expression of specific programs in specific cell types, ligand-receptor interactions etc. In Kasumi, it seems that a cell isn't necessarily assigned to a unique ME. At least from figure 2 it seems that the sliding widow analysis may assign a cell to different, overlapping MEs. How do the authors suggest that such data could then be incorporated in downstream analyses?

We agree that our approach significantly differs from and complements the standard spatial analysis workflow. We found it complementary, and in some contexts more informative, to identify conserved local and global spatial and molecular patterns across samples rather than assigning an individual label to each cell, as cells interact and are involved in complex spatial dependencies. In this way, we describe each image by the fraction covered by each persistent spatial dependency cluster. This does not impair biological interpretation and rather suggests multicellular processes underlying differences in clinical responses. This is illustrated by our new analysis of a pattern missed by all reference methods we tried, but encompassed by one Kasumi cluster, now explored in more detail in section 2.4 and Supplementary Figure S5. In the revised text we aimed to highlight a generally applicable workflow and how it can lead to insights into the molecular and cellular processes at play within samples. Further more targeted analyses such as cell-cell communication and ligand-receptor analyses can be performed following the same approach outlined in the manuscript, by defining appropriate views that capture processes of interest, where data is available. We now comment on this in the Discussion section.

“Nevertheless, while we considered view compositions capturing a single spatial context, Kasumi can be deployed with more complex compositions addressing

different spatial and functional contexts, different omics technologies, and alternative non-compositional representations at input akin recent applications of the global explainable relationship-based framework(MISTy).”

8. Adding running time analysis (including the running time of Misty) would be beneficial for the readers.

We added a subsection on Computational complexity of Kasumi in the Methods section of the revised manuscript. We outline the computational complexity per sample as a function of the number of windows, views and number of features. We derived the added factor of complexity of our local sliding window approach in contrast to a global run on the whole sample (MISTy). While the complexity increases linearly with the number of windows per sample, it also decreases log-linearly with the reduction of number of cells per window compared to the total number of cells.

We report that empirically “In our experimental setup, running on a newer generation laptop with 8 processing cores, the time taken per Kasumi run on a sample with cell-type information for each cell was $8.9(\pm 3.4)$ seconds. Running Kasumi on a sample with marker abundance level information was $30.5(\pm 24.9)$ seconds.“. We also added a supplementary figure showing the runtime per sample as a function of the number of cells and features per sample.

Note, also, that Kasumi is highly parallelizable on different levels. Each view-specific model for a target in each window and sample can be run in parallel. The underlying Random Forest models can also be parallelized, such that each tree in the ensemble is trained in parallel.

Minor comments:

- In the introduction, the authors state certain flaws in current approaches, such as relying on the underlying clustering of cell types and post-hoc ME annotation. This is not entirely correct, as many methods can now incorporate the underlying expression values. Moreover, Kasumi similarly works on top of cell types and requires post-hoc cluster annotation. In fact, utilizing cell types doesn't seem like a problem because when the authors switch to using proteins in figure 4, their performance in predicting clinical parameters mostly drops.

Using cell type information is a very common practice that is used by some related methods. We acknowledge that taking this step before performing more complex analyses simplifies the interpretation of results. However, there is no standard approach to cell type calling, which can sometimes lead to inconsistencies. We now list and compare to more approaches that work at the level of marker abundances.

In our analyses we show that for the tasks we defined Kasumi can deal with marker abundances at a similar level of performance as when using cell-type level information. The extended comparative analysis shows that while the performance of Kasumi slightly drops when compared to when using cell-type information it consistently outperforms the other related methods for clustering and neighborhood analysis at the level of marker abundances.

- The authors might consider rewriting, simplifying and shortening the abstract for clarity.

We removed redundant sentences and information from the abstract and added a definition of the concept of persistence as well as further clarifications of the introduced concepts.

- Regarding the equations in page 5:

o The analysis of the number of sliding windows in page 5 is trivial. It could be moved to the methods or supplement, as it's not very interesting or relevant for the average reader.

We moved the calculation of the number of windows to the Methods section in the context of calculating the computational complexity.

o The equations explaining the representation of window k are not clear.

We extended the description of the representation in the text following the equation in section 2.1. We also show the representations in Figure 1.

o Consider separating the text and the equations to different rows.

In the revised manuscript we separated the long equations from the text in sections 2.1, 4.2 and 4.6.

o Explain more clearly what each letter means.

The number of symbols in the equations in the main text is now significantly reduced. There is now a single equation in the Results section and every variable is explained.

o There is an overload of letter definitions in few lines. Consider simplifying the equations and/or moving them to the methods.

There is now only one equation describing the Kasumi model in the Results section, all other details are moved to the Methods section.

Reviewer #4 (Remarks to the Author):

We thank the Reviewer and express our support for the Nature Communications initiative for Early Career Researchers.

Response to reviewers

In the following, the reviewer's comments are shown in black while our responses are highlighted in blue. All text in quotation marks is taken from the revised manuscript. In the revised manuscript we also highlight the changes that resulted from addressing the comments in blue.

Reviewer #1 (Remarks to the Author):

In the revision, the authors made extensive efforts to improve their work. The novelty of using niche/local neighbors ("view" by the authors) to generate representation of spatial data remains unclear, as similar approaches exist (e.g., MENDER doi.org/10.1038/s41467-023-44367-9). The overall significance of this work requires further demonstration.

Despite the relatively simple algorithm, its description is unnecessarily lengthy (spanning four pages for Section 2.1) and is written with imprecise machine learning and mathematical language along with very limited biological interpretation. Notably, the revised version has further reduced the number of figures from five to four, which further diminishes the clarity and depth of the analysis. Several questions raised in my initial review were only partially addressed. My comments are focused primarily on Section 2.1 and authors' response letter. Significant works are needed for refinement and clarity of the manuscript.

1. (For comments 1-3 and 9-10) In Section 2.1, the description of "views" and the idea of "MISTy" seem to be clear. However, their description of "Kasumi" (their method), page 5-8, is still unclear. For example,

a. Why the two matrices in the right of Fig 1a have different type of column names in "predictor"?

In Section 2.1, page 5 we explain the approach to view construction in general, while the details of the view construction (especially intraview vs. paraview) related to the specific scenarios are explained in Section 2.2 and Section 2.4.

In Figure 1a and Supplementary Figure S1 the tables and heatmaps depict the representation and importances captured by the intraview (top) and the paraview (bottom). The paraview captures the broader tissue structure by aggregating representations in the neighborhood of each cell. For example, if the intraview captures the type of each cell the paraview captures the number of cells of a particular type of the 10 nearest neighbors of each cell. Therefore we represented the variables as a collection of cells of the same type.

We now revised Figure 1a and Supplementary Figure S1 such that the predictors are represented by the same symbol. We also added a description of the tables in the caption of Figure 1.

b. The paragraph “What is persistence, can it be visualized, quantified? Similarly, what is “similarity”?”

We revised Figure 1c to include a visual depiction of the persistence criterion. Similarity is described in Section 2.1 on page 7 as “To establish a similarity structure, we define and cluster a graph of similarities of all windows across all samples. Each node in the graph represents a window described by the Kasumi importance signature, i.e., the set of all estimated predictor-target relationships from that window with importance larger than zero (subsection Kasumi importance signatures). The nodes in the graph are connected by edges with weights equal to the cosine similarity between the node representations.”. We now updated the caption of Figure 1 with more specific information on the similarity.

c. (starting from first paragraph in page 7) The reduced/simplified version makes no improvement for clarity either in descriptions or mathematical formulation. For example, what is “meta-model”, “domain-specific functions”, “overlap o”, etc.. What are “i” and “j” for the importances matrices $M_{\{i,j\}}^{v,k}$? The authors describe index “i” as “target i” and “variable i in the intrinsic view” in the same equation, which is again confusing. The newly added paragraph “each sample is then represented by its compressed and relationship-oriented form of ...” can be deleted by simply describing the dimension of “M”.

In the revised text we now further specify the unclear terms. In particular: “ L_2 regularized late fusion linear meta-model that is trained on the predictions from the independently trained view-specific models”, “G are domain-specific functions that transform the intrinsic view based on spatial context X (for example, aggregation of the representation of the 10 nearest neighbors or distance-weighted sum of variables like like described in Section 2.4) and functional context T (for example, estimated pathway activities or selection of variables representing ligands or receptors)”, and “percentage of window overlap o”.

Note, also, that during modeling, each variable in the intraview is also considered as a target independently for model $Y_{.i}$. We use i consistently to represent a variable from the intraview that is also a target in this context. We now corrected the indexing mistake for M to $\{M_{\{j,i\}}^{v,k}\}$ and explicitly named the missing “predictors j in a window \$k\$” in the text. We later opted to describe the size of M verbosely since its dimensions are task and parameter specific.

- d. (Methods) The authors moved content to Methods to describe computational complexity. But nothing is related to clarify their method.
- e. In Section 2.1 and Methods, the authors need to find a way to present their overall ideas and the detailed algorithm in different levels.

We appreciate the constructive feedback from the reviewer and we hope that we now improved the clarity of our text by addressing the comments above. Section 2.1 of the manuscript is devoted to presenting our method in a comprehensive manner. To improve readability and as a result of the comments made by other reviewers in the previous round, we moved technical details previously in Section 2.1 to the Methods section. We also improved the implementation of the proposed method as a modular R package and added a vignette capturing the workflow presented in the manuscript, as a complementary and detailed representation of the algorithm at <https://github.com/jtanevski/kasumi>.

- f. The Random Forest utilizing “view” to predict “view” is a supervised model. Why the authors claim their method is unsupervised?

We apologize if this was not clear. The data-driven representation learning task is unsupervised, or rather self-supervised, while the Kasumi representations can be used for both unsupervised and supervised applications. For each window Kasumi trains models for each intraview variable as a target in a round robin manner. No additional data (label) is used in a supervised manner in any way. Independent models are trained for each view for the same target. While the models are trained by regression, they are not being used in a predictive or any other inductive setting. Instead, estimated feature importances are used to create a predictor-target-relationship representation. This can be seen as a self-supervised representation task, akin to training an autoencoder model and using the decoder weights as sample representation. Note, also, that based on this representation and central to Kasumi, the windows are clustered in an unsupervised manner. The representation learned by Kasumi can then be used downstream for any supervised or unsupervised task of interest. We added more details to reflect the above in Section 2.1 and Section 2.3 when referring to Kasumi as an unsupervised approach.

2. In Section 2.1, the authors now added definitions for “intraview” and “paraview” as two examples of “views”. Any other additional “views” were defined and used? Did the authors use the same set of “views” for all datasets tested in this work?

Yes, we use the same view composition consistently across all scenarios and datasets. Kasumi is able to also consider different view compositions to capture different types of interactions in different spatial contexts of interest. We opted for presenting the simplest composition, and making the workflow compatible across the different scenarios.

We modified the text of Section 2.1 to include a paragraph clarifying this.

“While the view composition can be defined flexibly and can be tailored to existing hypotheses or processes of interest, we focus here on a single-cell resolution and a composition of two views. In the following, all models consistently use the same view composition of an intraview and a paraview.”

3. (Reply to comment 3): “All of the information needed to generate the explanations is already available in the Kasumi output and can be obtained by backtracking the relationship...” What are explanation of the model? How to visualize, quantify and interpret the “explanation” is unclear. What kind of biological insights can be explained by the model is unclear.

In section 2.3 we present the different explanatory aspects of Kasumi in detail with particular focus on the task of distinguishing responders from non-responders in the CTCL data.

We first calculate and visualize the signed model reliance of each cluster in a predictive setting (Figure 3a, 4b and the newly added Supplementary Figure S11b), the distribution of the clusters across windows and samples (Figure 3c, 4c), and the significance of the spatial autocorrelation of highly relevant clusters (Figure 3d, 4d).

Next, each cluster of interest (here clusters that are associated with responders on non-responders) can be explained by two additional aspects of cluster-specific models. First, the amount of information gained by modeling the intraview and the information additionally gained when including the broader spatial context (paraview) per target. We visualize the latter in Figure 3b (top) and Supplementary Figure S7 as distribution of gain per target across all clusters from all samples for the four most important clusters. Second, relationships captured by the clusters by the particular cell types or markers that are of highest importance to predicting each target cell type or marker. These relationships, in light of the evidence of the performance of the cluster and its relevance to the predictive task, form the basis of the biological insights extracted with Kasumi (Figure 3b middle, Figure 4e top, and the newly added Supplementary Figure S11c). To improve the explainability we also provide an approximation of the sign of the relationship (colocalization or avoidance) by calculating the correlation between the predictor in its specific view and the target (Figure 3b, 4e bottom and Supplementary Figure S11c).

We further comment on the specific biological insights captured by the models and visualized in figures 3 and 4 in detail as well as make connections to the existing literature in sections 2.3 and 2.4.

4. (Reply to Comment 6): It is still unknown what does the “persistence” mean in the manuscript. The reply from the authors reads like the “persistence” basically is the “clusters” of all windows over different samples. What kind of analysis and findings were

discussed in this manuscript? Again, from my previous comment, what kind of property can this “persistence” provide?

We apologize that this is not clear yet. To further clarify this, in section 2.1 we have updated the description the persistence criterion which now reads:
“The existence of shared local structural (cell type) and functional (marker) patterns opens up a venue for exploring higher-level organization and empirical models of tissues. To define the building blocks of such representation as inputs to downstream tasks, as well as further reduce the complexity of the representation, we regularized the description by retaining only clusters that match a persistence criterion. Local patterns are meaningful for reasoning in terms of knowledge discovery and application to downstream tasks if they are persistently present across samples. The multiple independent observations of the same pattern within a subset of tissues in or across conditions adds to the relevance of the neighborhood to that tissue or condition. It also reduces the chances to include neighborhoods specific to a single sample that add noise and bias to downstream analyses. To this end we retain clusters that are present in at least 10% or at least 5 samples. The clusters not fulfilling this criterion are not considered further.”

The persistence criterion is indeed defined on the basis of the number of windows assigned a particular cluster label across all samples. The main purpose of the introduction of the persistence criterion is to remove sample specific clusters and non-abundant clusters that might introduce unnecessary patient or sample specific bias into the downstream analyses as well as regularize the representation. Our assumption is that in order for a pattern of cell-type or marker interactions to be deemed biologically meaningful, it should be observed in multiple instances.

Indeed, without the persistence criterion the performance of the representation across all tasks is significantly reduced. We added more information on the reduction of the number of features and performance in the main text. In particular, for the cell-type-based scenario in Section 2.2 “Not applying the persistence criterion leads to poorer results, despite a higher number of clusters (57, 28 and 28 for the DCIS, CTCL and BC datasets respectively). The performance reduced compared to the representation applying the persistence criterion (AUROC of 0.53, 0.66 and 0.53 compared to 0.79, 0.87 and 0.77 for the DCIS, CTCL and BC datasets respectively).” and for the marker-based scenario in Section 2.4 “Here, as in the cell-type-based scenario, bypassing the persistence criterion results in a representation based on a higher number of clusters and in a significant reduction in performance (AUROC of 0.63, 0.59 and 0.62 compared to 0.72, 0.78 and 0.77 for the DCIS, CTCL and BC datasets respectively).”

We additionally measure, visualise and comment on the properties of the distribution of the number of samples as a function of the number of windows where each cluster is present and the explicit cutoff imposed by the persistence criteria in Figure 3c and Figure 4c.

5. (Section 2.2) “By defining a global, non-spatial baseline we aim to show that there are no significant differences...” What is the baseline method?

In the cell-type-based scenario we take as baseline global representation the cell-type distributions per sample and in the marker-based-scenario the sum of abundances per marker relative to the total abundance in the sample.

We define the baselines towards the beginning of Section 2.2 and Section 2.4. We now updated the column names in the tables and added more details in the captions of Figure 2 and Figure 4.

6. The “views” can be either cell types or given markers. But the authors only test the case using cell types as views. However, markers usually provide more information and do not require cell type annotation than “cell types”. Shall the authors present the results like the explanation using “markers” than “cell types”?

We agree with the reviewer that markers should provide more information in general or more specific information on the underlying biological processes in the tissue. Therefore we aimed for Kasumi to be applicable on different types of representation. We further show that Kasumi can perform equally well and even better on marker-based input as on the cell-type-based input.

We consider both a scenario where cell types are given as input representation to Kasumi and a scenario where marker abundances per cell are given as input representation to Kasumi using data from the same datasets. In Section 2.4 we give a detailed overview of the specifics of the view construction, modeling approach and the performance of Kasumi in comparison to baselines and other related methods on marker based representations on the IMC breast cancer dataset. Similar as in the cell-type-based scenario we demonstrate how Kasumi is used to go from spatial measurements to biological insight. We detect and characterize relationship patterns between the markers in different spatial contexts within window clusters. We next identify clusters whose abundance is most relevantly associated with response to treatment, contextualize them based on their frequency and spatial distribution and comment in detail on the insights that were generated by this approach.

We updated the text of the Discussion section to include the following recommendation: “Our findings show that the availability of granular cell type information can lead to better performance. However, if only high level cell annotation is available then we expect that more information can be captured at the level of marker abundances. The same holds for measurements using technologies that are prone to contamination/dispersion of lineage markers to neighboring cells, preventing confident cell type identification. Since Kasumi is

scalable, it can be run with both representations at input and compare the performance of the output on a relevant downstream task.”

7. Why original Fig. 5 was removed?

The original Figure 5 was moved to the supplementary material and is now Supplementary Figure S8 to reduce technical details as per suggestions from the previous round of reviews. The references to the figure and the related observations remain in the text in Section 4.4.

Reviewer #2 (Remarks to the Author):

I thank the authors for the detailed responses. I now have a better understanding of your work. The revised manuscript is much easier to read and I really appreciate the addition of potential biological insights (fig 4e). I have a few follow up questions:

1) For clarity and completeness, can the authors complete the flowchart on Fig. 1, indicating how the persistent clusters are being used for the logistic regression models predicting responders vs non-responders and progressors vs. non-progressors? What are the size and value of the input matrix

In the revised manuscript we expanded Figure 1 to include a new panel d with more details on the Kasumi-cluster based representation and its use in a downstream task. It illustrates that each sample is represented by the distribution of persistent patterns. The input matrix for a downstream task is therefore $n \times c$ where n is the number of samples and c the number of persistent clusters. Given observations of the conditions or the outcome of the patient the representation can be used to train a logistic regression model to estimate how well the Kasumi representation captures these conditions and observations. The ability to use simple models based on a representation without further transformation suggests that the latter encodes relevant biological information and is easily interpretable. Finally, we use the insights from the reliance of the logistic regression model to facilitate the explanation by focusing on condition specific Kasumi clusters.

2) Why choose the value of 10 nearest neighbors? Have the authors tested other numbers? Should the number of neighbors be less than the sliding window size?

We chose to use a single value for the number of nearest neighbors for two reasons. First to capture the most similar concept of a neighborhood as the related methods. The concept of a neighborhood for CSEA and CCN is local and cell-centric, i.e., the immediate neighbors of each cell. In Risom et al. (DCIS data, CSEA) the cutoff radius for a neighboring cell is 50 μm and in

Phillips et al. (CTCL data, CCN) the 10 nearest neighbors. Second, we wanted to keep the views and modeling the same across all tasks for consistency and simplicity. Otherwise, as shown later for the quantitative analysis the paraview can be constructed in different ways and parameters, tailoring it to capture the spatial context of interest.

In Section 4.5 we report that “The size of the neighborhood and the number of clusters is chosen to match the information given in the original data and method papers.”.

The number of neighbors (or the parameter of the weighting function) should be chosen such that they are equal or smaller than the size of the window. In the revised manuscript in Section 2.1 we now note that “The Kasumi window clusters and the Kasumi representations are defined by the relationships in each window. As such, a Kasumi window with size smaller than the size of the paraview, capturing a group of cells with patterns beyond the scope of the window might result in low number or very similar clusters and misleading explanation.”

3) The estimated importance signature seems to serve as a kind of feature selection. Can the authors plot the importance score for one example dataset and show how many predictor-target relationships are one standard deviation above the mean importance per target and view?

For completeness of results we now show a heatmap with all interactions and their estimated importances to complement the filtered heatmaps shown in Figure 3 and Figure 4. We added a new Supplementary Figure S4 showing a complete set of predictor-target importances for the most relevant Kasumi clusters in the task of distinguishing responders from non-responders in the cutaneous T cell lymphoma data (cell-type scenario). We also added heatmaps with all predictor-target importances above zero for the most relevant clusters in the task of distinguishing responders from non responders in the breast cancer data (marker scenario) in panels b and d of Supplementary Figure S8. Shown are targets for which adding the paraview spatial context resulted in an increase of explained variance of more than 1. The other targets or importances related to those targets are not considered further in the Kasumi analysis. This also reflects the relationships comprising the window signatures for the first Kasumi representation.

4) The persistence criteria is a bit abstract and counter intuitive regarding how Kasumi can discover non-abundant relationships in the tissue. Can the authors show a plot from an actual data, how many and what are the clusters that did not pass the persistence criteria?

We revised Figure 1c to include a visual depiction of the persistence criterion for clarity. As we explain in more detail in Section 2.1 the persistence criterion is defined on the basis of the number of windows assigned a particular cluster label across all samples. The main purpose of the introduction of the persistence criterion is to remove sample specific clusters and non-abundant clusters that might introduce unnecessary patient or sample specific bias into the downstream analyses as well as regularize the representation. Our assumption is that in order for a pattern of cell-type or marker

interactions to be deemed biologically meaningful, it should be observed in multiple instances.

We added more information on the reduction of the number of features and performance in the main text. Without the persistence criterion the performance of the representation across all tasks is significantly reduced. In particular, for the cell-type-based scenario in Section 2.2 “Not including the persistence criterion results in a representation based on higher number of clusters (57, 28 and 28 for the DCIS, CTCL and BC datasets respectively) and a significant reduction in performance (AUROC of 0.53, 0.66 and 0.53 for the DCIS, CTCL and BC datasets respectively).” and for the marker-based scenario in Section 2.4 “Same as in the cell-type-based scenario not including the persistence criterion results in a representation based on higher number of clusters and a significant reduction in performance (AUC of 0.63, 0.59 and 0.62 for the DCIS, CTCL and BC datasets respectively).”

We measure, visualise and comment on the properties of the distribution of the number of samples as a function of the number of windows where each cluster is present and the explicit cutoff imposed by the persistence criteria in Figure 3c and Figure 4c.

The regions of the samples containing non-persistent clusters can be also visually appreciated first in Figure 2d as missing windows in the Kasumi representation compared to Figure 2c and further by comparing panels a and c in Supplementary Figures S2, S3, S5 and S6.

In the caption of Figure 2 we now make a note of this comparison as well as in the text of Section 2.2.

5) I really encourage the authors to demonstrate application in at least one spatial transcriptomics dataset. There are increasing number of high-resolution ST datasets with clinical phenotypes. For example, Shiau 2024 (DOI: 10.1038/s41588-024-01890-9) has SMI datasets for human PDAC tumors with (n = 6) or without (n = 7) neoadjuvant chemotherapy.

We agree that demonstrating Kasumi on different omics data improves the manuscript. We now demonstrate an application of Kasumi to the proposed dataset. We selected 300 samples (fields of view) coming from 12 patients. We removed samples from one treated patient that in addition to the capecitabine or 5-fluorouracil (CRT) treatment also received losartan treatment (CRTL). The scenario, unlike with the other datasets and tasks is a cross-condition one (treated vs. non-treated). As we now clarify in the Discussion section, in such data, the condition can be predicted without the need for spatial resolution or neighborhood analysis since simple baselines like the distribution of cell types perform extremely well (AUROC = 0.96). Therefore in the results sanction in the main text we opted for demonstrating Kasumi on data where global baselines are not sufficient to stratify the samples according to outcome. Nevertheless, applying Kasumi on the PDAC data resulted in slight improvement of the performance (AUROC = 0.97)

and a complementary spatial explanation consistent with the findings presented by Shiau et al. We report the results in the main text in the Discussion section. We also added a new figure with results from this application in the supplementary material (Supplementary Figure S11).

6) Discussion: do the authors have any recommendation or intuition as to which cell level representations of the samples, cell-type label or marker abundance level, is better? Should one always use the marker abundance mode.

We would recommend making a decision depending on the granularity and the confidence of phenotyping. If only high level cell annotation is available then we expect that more information can be captured at the level of marker abundances. The same holds for measurements using technologies that are prone to contamination/dispersion of lineage markers to neighboring cells, preventing confident cell type calling. Otherwise, as we observed in our benchmark, the availability of granular cell type information can lead to better performance. Since Kasumi is scalable, it can be run with both representations at input and compare the performance of the output on a relevant downstream task. For higher dimensional data (e.g. transcriptomics) we lean towards using either well captured cell states or representation based on functional enrichment of the data by estimation of the activities of higher level biological processes, such as pathways, or capturing a subset of relevant molecules like receptor expression in the interview and ligand expression in the paraview. We updated the text of the Discussion section to reflect these recommendations.

(Remarks on code availability)

I was able to install mistyR and kasumi. However, I found it challenging to reproduce the analyses shown in the main figures based on https://github.com/saezlab/kasumi_bench. Can the authors provide some vignettes (similar to <https://saezlab.github.io/mistyR/articles/>) and a Zenodo archive with scripts to reproduce the analyses in the paper?

We now include streamlined examples of the analysis in the examples folder at https://github.com/saezlab/kasumi_bench. In particular scripts to reproduce the analysis of the CTCL and the newly added PDAC data. The same analysis can be run in a similar way for all other data. There are small differences between the more monolithic implementation in our benchmark source code that was used to create the results for this manuscript and the cleaner and more modular package implementation. The examples follow the same analysis steps as in the benchmark code. We also deposited all source code together with the result databases and processed objects in Zenodo (<https://doi.org/10.5281/zenodo.14891956>) to ensure reproducibility of the results.

We improved the documentation of the package, added a vignette based on the modular implementation and step by step instructions for running the modular implementation of Kasumi on the CTCL data at <https://github.com/jtanevski/kasumi>. We are continuously developing the

package. As it matures and we encounter different applications of it we will also add new vignettes to it.

Reviewer #3 (Remarks to the Author):

Review for Tanevski et al.

The authors have addressed most of my comments. A few issues remain:

1. The authors added the window composition clustering as a baseline. However, both in the rebuttal and in the manuscript they refer to it as a novel idea (For example, “Our window composition clustering (WCC) approach” on page 15, or “such an approach can be useful to the community...” in the rebuttal). Performing window composition clustering is not novel. It is the baseline in the field. The authors should kindly modify the text in the paper to reflect this. Accordingly, they should rename the “Baseline” column in figures 2 and 4 to “Global composition”, since global composition is not really the baseline. WCC is the baseline.

To the best of our knowledge, there is currently no approach to analysis of spatially resolved multiplexed omics data at the level of segmented cells that implements a (overlapping) window based approach for the purpose of composition based or related clustering task. In fact, a recent implementation of a window composition based clustering approach in one of the broadly used libraries for analysis of spatial data was inspired by Kasumi (<https://github.com/scverse/squidpy/issues/829>).

In Figures 2 and 4 we renamed the baseline to global baseline and updated the figure legends to include the local composition nature of the WCC approach. We also updated the text to reflect that the WCC approach can be considered as a local composition based clustering baseline.

In Section 2.2 in the paragraph describing WCC we now note that “The resulting representation is comparable across windows and samples and can be further considered as a local composition-based baseline.”

2. Running times: The authors added running times to the methods section. However, running times seem to be very large, averaging 8-30 seconds per cell. For a normally-sized dataset of 1M cells, running Kasumi will take 3 months to a year... The discussion doesn't include a limitations paragraph. The authors should add it and it should clearly indicate running times.

We acknowledge that the Figure might have been a bit misleading in terms of what exactly we report. The reported times are for running Kasumi on a whole sample and not per cell. Each sample is represented by a single point in Supplementary Figure S10. The different samples also have a different number of cells. We now changed the axis labels to better capture this

information from cells vs time to cells per sample vs time per sample. We also updated the text to reflect this better. Running Kasumi on the newly added PDAC transcriptomics data took 85 minutes for processing all 300 samples and more than 700000 cells on a newer generation laptop with 8 processing cores.

Note, also, that Kasumi is run independently for each sample in the dataset. For large datasets with millions of cells across samples, the worst case runtime will be determined by the sample with the largest number of cells. Kasumi is designed with scaling in mind. Even for a sample with hundreds of thousands or millions of cells, Kasumi can be parallelized. For each sample, each window and even each view and target specific model can be run in parallel.

Reviewer #4 (Remarks to the Author):

We would like to thank the early career researcher again for their effort and constructive feedback to our work.

Response to reviewers

In the following, the reviewer's comments are shown in black while our responses are highlighted in blue. All text in quotation marks is taken from the revised manuscript.

Reviewer #1 (Remarks to the Author):

No further comments.

We would like to thank Reviewer #1 for their constructive feedback during the review process.

Reviewer #2 (Remarks to the Author):

The authors have answered all my questions. I also want to thank the authors for providing the example vignette. I have a minor comment about Supplementary Figure S10: It is a useful plot, but very hard to read. Consider changing from yellow to a darker color, try log scale for the y axis?, and please indicate which samples are being plotted. I have no other questions.

We would like to thank Reviewer #2 for their constructive feedback during the review process.

In Supplementary Figure S10 we now changed the yellow color to a darker (brown) one. We also log-scaled the y-axis and for each sample in the figure legend we now include information from which datasets it comes from. We also report exact times, number of cells per sample, number of features and corresponding dataset for each point in the supplementary Source Data file.

Reviewer #3 (Remarks to the Author):

Review 3 for Tanevski et al.

1. I disagree. Almost every spatial proteomics paper that I know of used clustering in a small window to define microenvironments. Examples include (but are not limited to): Goltsev et al., Cell 2018 ; Schurch et al., Cell 2020 ; Hoch et al., Science Immunology 2021. I remain adamant in my previous request, not addressed by the authors:

We understand the perspective of the reviewer and we would like to further clarify the differences between the cell-centric and pairwise-association based approaches proposed in the references above and the window-centered approach we are proposing.

We reference and comment on the first two papers and the approaches described in them. Namely iNiche and Coordinated Cellular Neighborhoods - “Another group, consisting of methods such as iNiche[16], Spatial-LDA[17], and Coordinated Cellular Neighborhoods (CCN)[18,19], identify neighborhood motifs by first representing each cell by the cell-type composition within its neighborhood and then clustering this representation again to infer higher-order motif-oriented identities.”.

The CCN-like approaches consider as “window” the immediate neighborhood (k spatially nearest neighbors) centered on each cell in the sample. The representation they are generating is therefore a label at the level of each cell. Kasumi in contrast is not centered on a specific cell but defines a window covering a specific region. The representation generated by Kasumi is a label at the level of each window.

Hoch et al. for neighborhood analysis reference the procedure taken in Schulz et al. Cell Systems 2018, which in turn reports the use of histoCAT (Schapiro et al. Nature Methods 2017, citation 12 in our manuscript) which we cite a part of the group of methods that take as an approach the estimation of pairwise association of cell types - “Methods for neighborhood analysis adopt different approaches. Some, such as histoCAT[12], Giotto[13], and Cellular Spatial Enrichment Analysis (CSEA)[14,15], focus on the immediate neighborhood and number of interactions between pairs of cell types or functional states. The pairs of interacting cell types are identified by calculating the significance of co-occurrence within the immediate neighborhood by comparing to a null distribution of interactions derived from cell location permutations.” We refer to this group of methods by one of its representatives, CSEA.

The CSEA-like approaches generate a representation based on the statistics of pairwise colocalization and association in the immediate neighborhood at the level of the whole sample.

We compare the performance of Kasumi and WCC to these related approaches (named as CCN and CSEA) in the Results section.

a. That the authors remove statements of novelty of the WCC approach.

While both in terms of input and the level of representation the WCC approach is different from the other approaches as outlined above, we don’t explicitly claim WCC as a particular novelty point of our work. We report on WCC as an approach that we implemented to generate a window-level representation that is most compatible with the output of Kasumi. We removed all mentions of the terms new/novel from the manuscript.

b. That the authors rename “Global baseline” to “Global composition”. Like I said in the previous comments, this is not in any way the baseline in the field.

I believe that these requests are extremely reasonable and easy to implement.

To clarify this point, we now renamed the global baseline to global composition. While currently the global composition might not be always considered as a baseline, our results show that for

translational tasks the non-spatial global composition (cell types and markers) outperforms other spatial-based representations. For other tasks, such as distinguishing between treated and not treated samples (Supplementary Figure S11) it performs almost perfectly, which can also be the case for other routine clinical observations where spatial omics should not be necessarily indicated.

2. The authors have addressed my comment.

We would like to thank Reviewer #3 for their constructive feedback during the review process.

Reviewer #4 (Remarks to the Author):

We would like to thank the early career researcher again for their effort and constructive feedback to our work during the review process.